



# Soil moisture and precipitation intensity control the transit time distribution of quick flow in a flashy headwater catchment

Hatice Türk[1], Christine Stumpp[1], Markus Hrachowitz[2], Karsten Schulz[3] Peter Strauss[4], Günter Blöschl[5], and Michael Stockinger[1]

[1]University of Natural Resources and Life Sciences, Vienna, Department of Water, Atmosphere and Environment, Institute of Soil Physics and Rural Water Management, Muthgasse 18, 1190 Vienna, Austria
[2]Department of Water Management, Faculty of Civil Engineering and Geosciences, Delft University of Technology, Stevinweg 1, 2628CN Delft, the Netherlands
[3]University of Natural Resources and Life Sciences, Vienna, Department of Water, Atmosphere and Environment, Institute of Hydrology and Water Management, Muthgasse 18, 1190 Vienna, Austria
[4]Institute for Land and Water Management Research, Federal Agency for Water Management, Petzenkirchen, Austria
[5]Vienna University of Technology, Institute of Hydraulic Engineering and Water Resources Management, Karlsplatz 13, 1040 Vienna, Austria

Correspondence to: Hatice Türk, (hatice.tuerk@boku.ac.at)

**Abstract.** The rainfall-runoff transformation in catchments usually follows a variety of slower and faster flow paths which leads to a mixture of "younger" and "older" water in streamflow. Previous studies have investigated the time-variable distribution of water ages in streamflow (Transit Time Distribution, TTD) by stable isotopes of water ($\delta^{18}O$, $\delta^{2}H$) together with transport models based on StorAge Selection (SAS) functions. This function traditionally formulated based on soil moisture to mimic preferential release of younger water as the system becomes wetter. However, besides soil moisture, it is plausible to assume that precipitation intensity may also play a critical role in how quickly water flows through a catchment. In this study, we tested whether fast flow and its transit times are controlled by soil moisture only or also by precipitation intensity in a heterogeneous catchments with a significant fast runoff response component. We analyse high-resolution δ18O data (weekly and event streamflow δ18O samples) in a 66 ha agricultural catchment. We estimate TTDs by a tracer transport model based on SAS functions. We test two scenarios of the SAS function parameter for the quick release of young water into streamflow, one as a function of soil moisture only, and one as a function of both soil moisture and precipitation intensity. The results that accounting for both soil moisture and precipitation intensity to define the shape of SAS functions for quick flow, improved the tracer simulation in streamflow (increase in Nash-Sutcliffe Efficiency from 0.31 to 0.51). Even though the estimation of the TTs younger than 90 days were similar for both SAS approaches, the shorter travel times(TTs younger than 7 days) were not represented well when only accounting for soil moisture in the SAS function parameterization, in particular, in the summer and autumn months. This is due to flow processes that promote the direct contribution of precipitation to the stream (e.g tile drain) and infiltration excess overland flow processes. It appears that a significant portion of event water bypasses the soil matrix through fast flow paths (overland flow, tile drains, and/or preferential flow paths) also in dry soil condition for both low and high-intensity precipitation. Thus, in catchments where preferential flows and overland flow are important flow processes, soil-wetness-dependent and precipitant-intensity-conditional SAS functions may be required to better describe and identify the mechanisms behind the quick streamflow generation and their time scale.





## 1 Introduction

The focus of hydrological research has expanded from the quantitative estimation of water fluxes to better descriptions of underlying hydrological processes by estimating the water age of various storage and runoff components in catchments (Beven , 2006; McDonnell & Beven, 2014; Sprenger et al., 2019). Water age can give crucial information about the pathways through which water moves in catchments and their dynamics. This information can help to identify the partitioning of precipitation into distinct fluxes such as overland flow, lateral subsurface flow and deep percolation. These are useful for understanding the fate of pollutants and sediments which is essential in managing water resources sustainably.

The time it takes for precipitation to reach the stream is referred to as water transit time, while water age is the time that has elapsed since precipitation entered the catchment (Rinaldo et al., 2011; Botter et al., 2011; Benettin et al., 2022). The age distribution of water stored in the catchment is referred to as the residence time distribution (RTD). Depending on a catchment's physical characteristics and on hydrometeorological conditions, transit times may vary between seconds and decades. Therefore, the Transit Time Distribution (TTD) is essential for representing transport processes in catchments (McGuire & McDonnell, 2006; Botter et al., 2011; Klaus & McDonnell, 2013; Benettin et al., 2022).

Conservative environmental tracers, such as the stable isotopes of oxygen ($\delta^{18}$O) and hydrogen ($\delta^{2}$H) in water, have been widely used to investigate water ages as well as runoff generation processes (Kirchner et al., 2000; Fenicia et al., 2008; McGuire & McDonnell, 2006; Klaus & McDonnell, 2013; Wang et al., 2023) and their time variance (Fenicia et al., 2010; McDonnell & Beven, 2014; Benettin et al., 2022; Wang et al., 2024). These tracers play a critical role in estimating where, how, and how quickly water is mobilized from the landscape, especially for quantifying water age distributions along surface and subsurface flow paths (McDonnell & Beven, 2014; Sprenger et al., 2019).

Recent developments in sampling techniques have improved the spatiotemporal resolution of the measured stable isotope data, e.g., hourly $\delta^{18}$O of precipitation (von Freyberg et al., 2022; Welb et al., 2022), or sub-daily to daily $\delta^{18}$O of streamflow (von Freyberg et al., 2022; Dahlke et al., 2014). This has improved our ability to track the partitioning of precipitation into different hydrological fluxes such as root water uptake, plant transpiration, overland flow, lateral subsurface flow, groundwater recharge, and eventually streamflow (Hrachowitz et al., 2015; Abbott et al., 2016; Knighton et al., 2019; Knighton et al.,2020; Kübert et al., 2023). Tracer-aided hydrological models have been developed and made it possible to investigate the contributions of distinct runoff generation mechanisms, such as overland flow or groundwater flow by solving water-, tracer- and associated water age balances (Botter et al., 2011) to estimate water transit times (Hrachowitz et al., 2013; Benettin et al., 2015; Lutz et al., 2018; Kuppel et al., 2018; Remondi et al., 2019; Wang et al., 2023, 2024).

Recent advances in TTD estimation have improved our ability to describe the relationship between storage and discharge in hydrological systems using the StorAge selection (SAS) approach (Botter et al., 2011; Rinaldo et al., 2015). The SAS function describes the probability with which water parcels of different age in a catchment's storage are released, therefore representing the relative contribution of young and old water to streamflow (Botter et al., 2011; Rinaldo et al., 2015). However, SAS functions cannot be directly observed. Instead, they are



typically inferred from calibration of a tracer-aided hydrological model that fits modelled tracer and streamflow signals to observed ones. They can be defined either as time-variable or -invariable functions (Hrachowitz et al., 2013) with various functional shapes, such as beta (van der Velde et al., 2012), Dirac delta (Harman, 2015) or
80 gamma (Harman, 2015) distributions.

Previous studies showed that soil moisture (soil storage) is a controlling factor for the time-variable shape of the SAS function, thus accounting for the higher probability of the release of young water as a catchment's soil wets up (Harman, 2015; Hrachowitz et al., 2016; Benettin et al., 2017; Kaandorp et al., 2018; Harman, 2019). This is sometimes also referred to as "inverse storage effect" (Harman, 2015). The wetness-dependent time variability
of SAS functions was implemented in hydrological models to simulate the tracer fluctuations in streamflow in catchments, such as Claduègne (Hachgenei et al., 2024), Gårdsjön (van der Velde et al., 2015), Elsbeek and Springendalse Beek (Kaandorp et al., 2018) Plynlimon (Benettin, et al., 2015; Harman, 2015) and several Scottish catchments (Hrachowitz et al., 2013). Time-variable parameterization of the SAS function depending on catchment wetness may be needed in catchments due to various factors. These factors include: (i) the
dominance of a single process dependent on soil moisture conditions like Hafren catchment in Wales (Benettin, et al., 2015; Harman, 2015) or saturation-excess overland flow in the Bruntland Burn catchment in Scotland (Benettin et al., 2017a) (ii) other site-specific hydrological characteristics that may be primarily influenced by catchment wetness.

However, exclusively basing the shape of the SAS function on soil moisture may not fully capture the
95 complexity of hydrological responses in catchments. Danesh-Yazdi et al. (2018) and Rodriguez and Klaus (2019) suggest that such a parameterization of SAS functions based on storage, and thus representing (soil) wetness, may not capture all relevant transport processes due to nonlinear relationships between storage and streamflow as observed in catchments like the flashy Weierbach in Luxembourg (Rodriguez and Klaus, 2019) and the Hydrological Open Air Laboratory in Austria (Vreugdenhil et al., 2022). This may in particular be true for
catchments with moderate to low infiltration capacity of soils, where the intensity and duration of precipitation can also play a critical role in how quickly water is mobilized from the landscape (Blöschl, G. 2022).

Headwater catchments are often characterized by quick flow processes, such as overland flow and preferential flow through macropores in the shallow subsurface (Weiler and McDonnell, 2007; Klaus et al., 2013; Angermann et al., 2017; Loritz et al., 2017; Maier et al., 2021). In transport models, this preferential flow
process is implicitly encapsulated in the SAS functions (Hrachowitz et al., 2021). So far, the rapid response has therefore been mostly considered as preferential flow or as saturation excess overland flow as function of soil moisture. However, a rapid response can also occur when rainfall intensities exceed the infiltration capacity (i.e. Hortonian runoff generation). Therefore, it remains to be tested whether accounting for precipitation intensity in addition to soil moisture to parameterize time-variable SAS functions may yield improved representations of
stream tracer dynamics in specific environments.

The main objective of this study was to test two different approaches to determine the shape of time-variable SAS functions for fast runoff generation in a flashy headwater catchment: (i) soil moisture alone controls the





SAS function shape for quick flows, and (ii) soil moisture and precipitation intensity jointly control the SAS function shape for quick flows.

## 2 Methodology

### 2.1 Study Site

The Hydrological Open-Air Laboratory (HOAL) is a 66-hectare site located in Petzenkirchen, Austria (Blöschl et al., 2016) (Fig. 1). The catchment is characterized by a humid climate with an average annual air temperature of 9.5°C. The mean annual precipitation and runoff are 823 mm yr$^{-1}$ and 195 mm yr$^{-1}$, respectively. The year 2015 was notably dry (P = 580 mm yr$^{-1}$) while 2013, 2014, 2016, and 2017 had higher precipitation levels (> 700 mm yr$^{-1}$) and were classified as relatively wet years. The elevation is between 268 and 323 m above sea level, with an average terrain slope of 8%. The predominant soil types are Cambisols (57%), Planosols (21%), Kolluvisol (16%), and Gleysols (6%). The soils are characterized by a high clay content of 20–30% (Blöschl et al., 2016; Eder et al., 2014). Land use primarily includes agriculture (87%) (crop cultivation of maize, winter wheat, rape and barley), forest (6%), pasture (5%) and paved areas (2%) (Blöschl et al., 2016). The concave part of the catchment (Fig. 1) was tile drained in the 1940s to reduce water logging because of the shallow, low-permeability soils and the catchment's use as agricultural land. The estimated drainage area from the tile drains is about 15% of the total catchment area (Fig. 1).

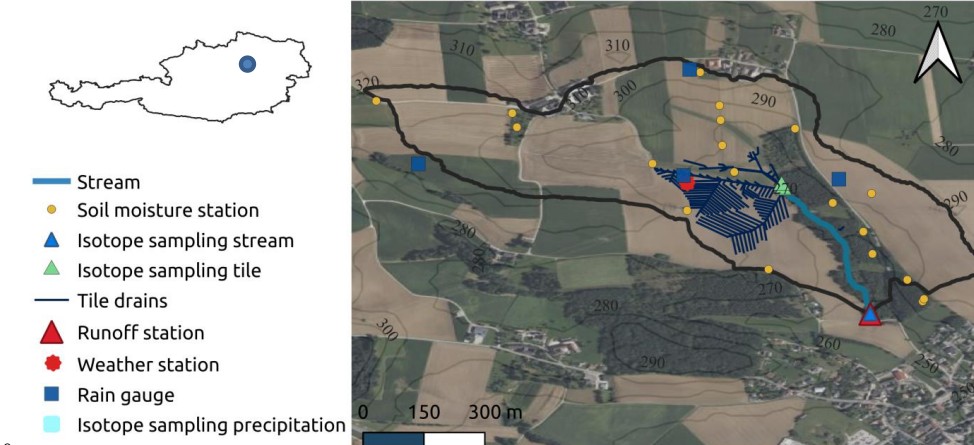

**Figure 1.** Map of the HOAL catchment (66 ha, Lower Austria) and location of devices for precipitation, weather station, soil moisture, isotope sampling from stream, and isotope sampling from precipitation (located approximately 300 m south of the catchment, light blue circle)(map image from © Microsoft, Bing Maps via Virtual Earth )

### 2.1.1 Hydrometeorological data

Hydro-meteorological data for the time period between October 2013 and 30 December 2018 were used for the analyses (Fig. 2a). For this time period, daily precipitation was available from four weighing rain gauges (OTT Pluvio) (Fig. 1). The arithmetic mean of the four rain gauges was here in the following used as catchment



average precipitation (Fig. 2a). Daily runoff at the catchment outlet was monitored using a calibrated H-flume
with a pressure transducer (Fig. 2a). Daily soil moisture in the unsaturated zone was available through 19
permanent (Fig. 1). For this study, the catchment average soil water content was calculated across four different
depths: 0.05 m, 0.10 m, 0.20 m, and 0.50 m and used for the analyses.. Sensor specifications and additional
details about the hydrometeorological data are provided in Blöschl et al. (2016).

### 2.1.2 Stable isotope data

$\delta^{18}$O measurements for the time period between October 2013 and 30 December 2018 were used for the
analyses (Fig. 2b-d). During this time period, precipitation isotope samples (Fig. 2b) were collected using an
adapted Manning S-4040 automatic sampler, located approximately 300 meters south of the catchment (Fig. 1).
This sampler, coupled with a rain gauge, collected water after every 5 mm. If the events intensities were less
than 5mm, the mixing of precipitation at the end of with that of the following event can occur. For this events
the average concentration of temporally separated events were used. In addition to weekly grab samples (Fig.
2d), discharge water at the catchment outlet was collected during precipitation events using an Isco 6712
automatic sampler for the period from 2013 onwards (Fig. 2c). Similar to discharge, water samples were
collected at the outlet of tile drains at two location (Fig. 1) during precipitation events using an Isco 6712
automatic sampler. Sample collection for stream and tile drain water was based on specific flow rate thresholds,
varying the sampling frequency from 15 minutes to 2 hours depending on the anticipated length of the event.
Analysis of these water samples for the stable isotopes of oxygen ($^{18}$O/$^{16}$O) and hydrogen ($^{2}$H/$^{1}$H) was done
using Picarro L2130-i and L2140-i laser spectrometers (cavity ring-down spectroscopy). The measurement
uncertainties were ±0.1‰ for $\delta^{18}$O and ±1.0‰ for $\delta^{2}$H, respectively. All isotopic measurements are reported in
per mil (‰) relative to Vienna Standard Mean Ocean Water (VSMOW). Both precipitation and streamflow
event samples, as well as tile drainage samples, were aggregated to daily time intervals by calculating the
volume-weighted average of the sampling fluxes based on their sampling frequency.



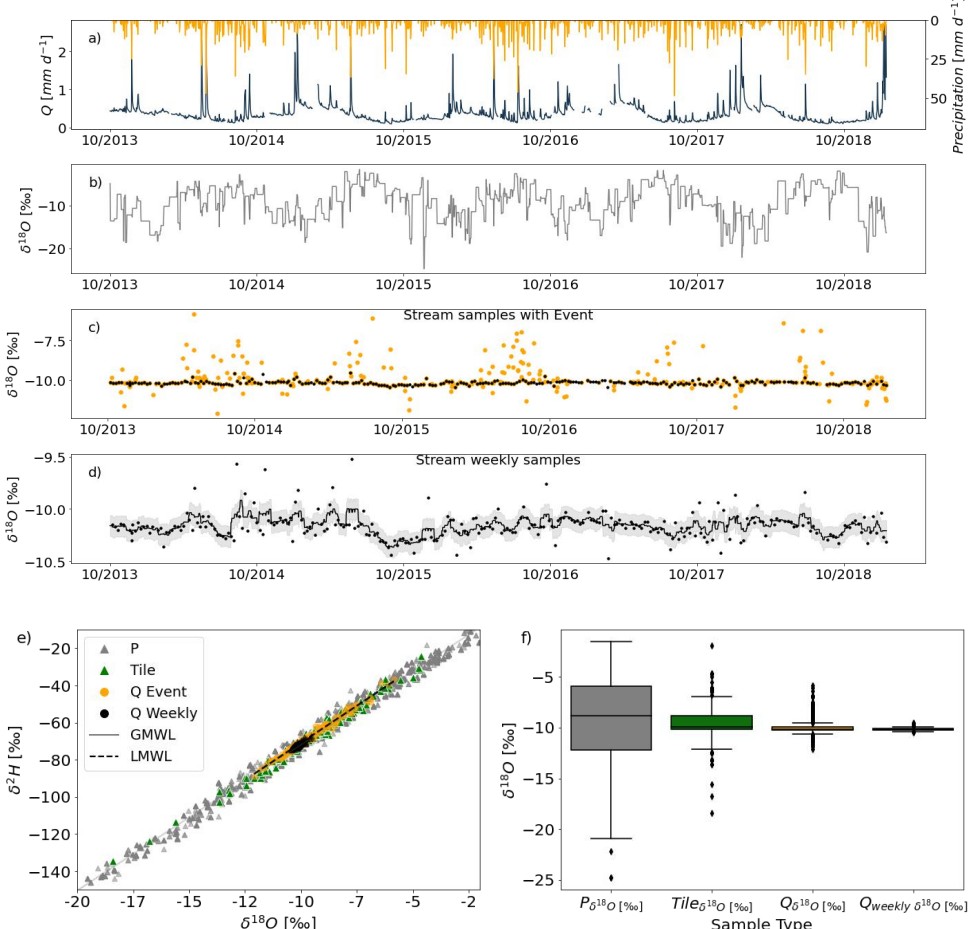

**Figure 2.** Hydrological and tracer data of the HOAL catchment (a) daily observed precipitation P (mm d⁻¹) and stream flow Q (mm d⁻¹) (b) δ¹⁸O data from precipitation event samples at daily time scale (c) δ¹⁸O data from streamflow with event (orange) and weekly grab samples (black) (d) weekly δ¹⁸O data from streamflow where the gray shaded area shows the measurement uncertainty of ± 0.1‰ (e) dual plot of δ¹⁸O and δ²H from precipitation event samples (grey dots), streamflow event samples at daily time scale (orange dots), weekly grab samples (black) and tile drain event samples at daily time scale (green) (f) Box plot of δ¹⁸O signal from precipitation event samples (gray box), tile drainage (green box), stream flow event (orange box) and weekly (black box).





### 2.2 Hydrological model structure

The process-based model used in this study consists of five reservoirs based on the previously developed DYNAMITE modeling framework (Hrachowitz et al., 2014; Fovet et al., 2015). The reservoirs represent the

storage components for snow ($S_{snow}$, Eq. 1), canopy interception ($S_i$, Eq. 2), unsaturated root zone ($S_r$, Eq. 3), fast response ($S_f$, Eq. 4) and groundwater with active and passive components ($S_{S,a}$ and $S_{S,p}$, Eq. 5). Each of these had its own associated water fluxes (Fig. 3). The water balance and flux equations of the individual model components are given in Table 1 and a complete list of parameters and their upper and lower bounds can be found in Table 2. A detailed model description and rationale for the assumptions in the model architecture can

be found in previous studies (Hrachowitz et al., 2014; Fovet et al., 2015).

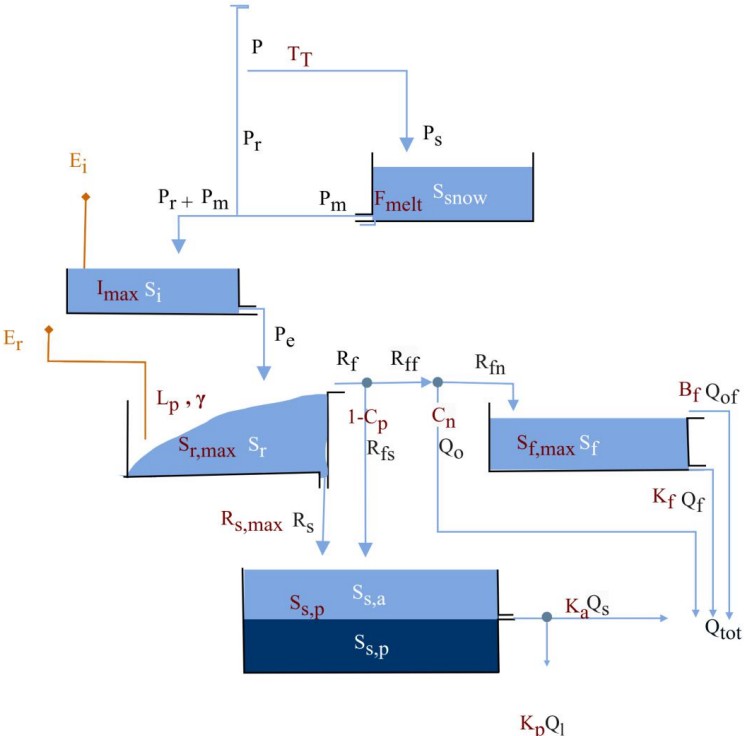

**Figure 3.** The model structure used to represent the HOAL catchment. Light blue boxes indicate the hydrologically active individual storage volumes that contribute to total discharge $Q_{tot}$: Snow storage ($S_{snow}$), canopy interception ($S_i$), fast response bucket ($S_f$), root zone ($S_r$), and "active" groundwater ($S_{s,a}$). The darker blue box $S_{s,p}$ indicates a

hydrologically "passive" mixing groundwater volume. Blue lines indicate snow and water fluxes while the orange lines indicate water vapor fluxes. Model parameters are shown in red adjacent to the model component they are associated with. All symbols are defined in Table1 and Table 2.





**Table 1:** Water balance and constitutive equations of the hydrological model (Fig. 3). P (mm d⁻¹) is total precipitation, $P_s$
(mm d⁻¹) is solid precipitation (snow), $P_r$ (mm d⁻¹) is liquid precipitation (i.e. rain), $P_m$ (mm d⁻¹) is snowmelt, $P_e$ (mm d⁻¹) is
throughfall, $E_i$ (mm d⁻¹) is interception evaporation, $E_a$ (mm d⁻¹) is evaporation from the root zone, $R_f$ (mm d⁻¹) is total
preferential fast response, $R_{fs}$ (mm d⁻¹) is fast recharge to slow-responding reservoir, $R_{ff}$ (mm d⁻¹) preferential fast response,
$Q_o$ (mm d⁻¹) is infiltration excess overland flow, $R_{fn}$ (mm d⁻¹) is preferential fast response to the fast-responding bucket, $Q_f$
(mm d⁻¹) is flow from the fast-responding reservoir, $Q_{of}$ (mm d⁻¹) is saturation excess overland flow from the fast-response
bucket, $R_s$ (mm d⁻¹) is slow recharge to slow-responding reservoir, $Q_s$ (mm d⁻¹) is flow from the slow-responding reservoir,
$Q_l$ (mm d⁻¹) is deep infiltration loss and $Q_{tot}$ (mm d⁻¹) is the total discharge. A list of model parameters and their definitions
are provided in Table 2.

| Storage Component and Water Balance | Eq. | Constitutive equations | Eq. |
|---|---|---|---|
| **SnowBucket** $$\frac{dS_{snow}}{dt} = P_s - P_m$$ | (1) | $$P_s = \begin{cases} P, & T < T_T \\ 0, & T \geq T_T \end{cases}$$ | (6) |
| | | $$P_m = \begin{cases} 0, & T < T_T \\ min(F_{melt}(T - T_T), \frac{S_{snow}}{dt}), & T \geq T_T \end{cases}$$ | (7) |
| **Interception storage** $$\frac{dS_i}{dt} = P_r + P_m - P_e - E_i$$ | (2) | $$P_r = \begin{cases} 0, & T < T_T \\ P, & T \geq T_T \end{cases}$$ | (8) |
| | | $$P_e = max(0, \frac{S_i - I_{max}}{dt})$$ | (9) |
| | | $$E_i = min(E_p, \frac{S_i - I_{max}}{dt})$$ | (10) |
| **Soil storage** $$\frac{dS_r}{dt} = P_e - R_f - R_s - E_a$$ | (3) | $$Cap = (1 + \gamma)S_{r,max}\left(1 - max(0, (1 - \frac{S_r}{S_{r,max}}))^{\frac{1}{1+\gamma}}\right)$$ | (11) |
| | | $$R_f = P_e - S_{r,max} + S_r + S_{r,max}\left(1 - \frac{(P_e + Cap)}{(1 + \gamma)S_{r,max}}\right)^{(1+\gamma)}$$ | (12) |
| | | $$R_s = min(R_{s,max}\frac{S_r}{S_{r,max}}, \frac{S_r}{dt})$$ | (13) |
| | | $$E_a = min\left((E_p - E_i)min(\frac{S_r}{S_{r,max}L_p}, 1), \frac{S_r}{dt}\right)$$ | (14) |
| **Division fast recharge and fast flow and overland flow** | | $$Rff = (C_p)R_f$$ | (15) |
| | | $$R_{fs} = (1 - Cp)R_f$$ | (16) |
| | | $$Q_o = \begin{cases} 0, & P_r < P_{tresh} \\ CnR_{ff}, & P_r \geq P_{tresh} \end{cases}$$ | (17) |
| | | $$Rfn = (1 - C_n)R_{ff}$$ | (18) |
| **Fast responding Bucket** $$\frac{dS_f}{dt} = R_{fn} - Q_{of} - Q_f$$ | (4) | $$Q_{Of} = max((S_f(\frac{S_f}{S_{f,max}})^{B_f} - S_{f,max}), 0)$$ | (19) |
| | | $$Q_f = max(0, (S_f(1 - exp^{(-k_f t)})))$$ | (20) |



$$S_{s,tot} = S_{s,a} + S_{s,p} + R_s + R_{fs} \qquad (22)$$

$$Q_{s,tot} = \frac{S_{s,tot} - S_{s,tot,out}}{dt} \qquad (23)$$

**Groundwater storage**

$$\frac{dS_{s,a}}{dt} = R_s + R_{fs} - Q_s - Q_l \qquad (5)$$

$$\frac{Q_s}{Q_l} = max(0, \frac{k_a(S_{s,tot} - S_{s,p})}{k_p S_{S,tot}}) \qquad (24)$$

$$Q_s = \frac{\frac{Q_s}{Q_l} Q_{s,tot}}{(\frac{Q_s}{Q_l} + 1)} \qquad (25)$$

$$Q_l = \frac{Q_{s,tot}}{(\frac{Q_s}{Q_l} + 1)} \qquad (26)$$

**Table 2**: Definitions and uniform prior distributions of the parameters of the solute-transport model (Fig. 3)

| Parameter | Unit | Definition | Lower Bound , Upper Bound | Calibrated S1,S2 |
|---|---|---|---|---|
| **Hydrological** | | | | |
| $T_T$ | (°C) | Threshold temperature for snow melt | [-4.0, 5.0] | [-2.90, -3.25] |
| $\gamma$ | (−) | Shape factor | [0.0, 5.0] | [0.09, 0.19] |
| $B_f$ | (−) | Saturation excess overland flow coefficient | [0.0, 0.00001] | [7.39e-6, 4.06e-06] |
| $C_n$ | (−) | Division parameter for fraction of overland flow | [0.0, 1.0] | [0.33, 0.18] |
| $C_p$ | (−) | Division parameter for fast groundwater recharge | [0.0, 1.0] | [0.36, 0.28] |
| $F_{mel}t$ | (mmd$^{-1}$ °C$^{-1}$) | Melt factor | [1.0, 5.0] | [2.14, 1.65] |
| $I_{max}$ | (mm) | Interception capacity | [1.2, 5.0] | [1.23, 1.82] |
| $K_a$ | (d$^{-1}$) | Storage coefficient of the slow-responding reservoir | [0.01, 1.2] | [0.19, 0.20] |
| $K_f$ | (d$^{-1}$) | Storage coefficient of the fast-responding reservoir | [0.01, 2.0] | [1.24, 0.85] |
| $K_p$ | (d$^{-1}$) | Storage coefficient of deep infiltration losses | [0.0, 0.00001] | [1e-05, 1e-04] |
| $L_p$ | (−) | Transpiration water stress factor | [0.0, 1.0] | [0.55, 0.387] |
| $P_{tresh}$ | (mm d$^{-1}$) | Threshold precipitation for overland flow | [2.0, 20.0] | [9.92, 6.25] |
| $R_{s,max}$ | (mm d$^{-1}$) | Maximum percolation rate | [0.0, 1.2] | [0.61, 0.63] |



| $S_{f,max}$ | (mm) | Fast response storage capacity | [0.0, 20.0] | [6.34, 4.25] |
|---|---|---|---|---|
| $S_{r,max}$ | (mm) | Root-zone storage capacity | [100, 500] | [285, 382] |
| **Tracer** | | | | **Tracer** |
| $S_{S,p}$ | (mm) | Passive storage capacity | [1000, 10000] | [7555, 3173] |
| $S_{U\_Alpha}$ | (−) | SAS alpha shape parameter for root zone | [0.00, 1.0] | [0.03 0.06] |
| $S_{G\_Alpha}$ | (−) | SAS alpha shape parameter for GW | [0.98, 1.0] | [0.99, 0.99] |


Precipitation $P$ (mm d$^{-1}$) below the threshold temperature $T_T$ (°C) enters the catchment as snow $P_s$ (mm d$^{-1}$, Eq.6) and accumulates in the snow bucket $S_{snow}$ (mm). Snowmelt $P_{m,}$ (mm d$^{-1}$) is then computed with the degree-day method (Eq. 7), driven by the melt factor $F_{melt}$ (mm d$^{-1}$ °C$^{-1}$) as described by Gao et al. (2017) and Girons Lopez et al. (2020). Rainwater $P_r$ (mm d−1), combined with snow melt $P_m$ (mm d$^{-1}$) passes through the canopy

interception storage $S_i$ (mm). Water that is not evaporated as interception evaporation $E_i$ (mm d$^{-1}$, Eq. 10) enters the unsaturated root zone $S_r$ (mm) as throughfall $P_e$ (mm d$^{-1}$, Eq. 9) based on the water balance of canopy interception storage (Nijzink et al., 2016) (Eq. 2). Water from the root zone $S_r$ (mm) can either be released as (i) fast discharge $R_f$ (mm d$^{-1}$, Eq. 12), which is based on a critical storage capacity Cap calculated using $S_{r,max}$ and the shape factor $\gamma$ (-) (ii) slow recharge to the active groundwater storage $S_{s,a}$ (mm) through a slower percolation

flux $R_s$ (mm d$^{-1}$, Eq. 13) which is driven by the maximum percolation rate $R_{s,max}$ (mm d$^{-1}$) (iii) the combined flux of root-zone transpiration and soil evaporation $E_a$ (mm d$^{-1}$, Eq. 14) defined by the transpiration water stress factor $L_p$ (−). The fast, preferential discharge $R_f$ (mm d$^{-1}$) is subsequently divided in several steps to account for fast flow paths. These are the preferential flow recharging groundwater $R_{fs}$ (mm d$^{-1}$, Eq. 15), the infiltration-excess overland flow reaching streamflow $Q_o$ (mm d$^{-1}$, Eq. 16) which is regularly observed in the HOAL

catchment (Blöschl et al., 2016) and the lateral subsurface flux $R_{fn}$ (mm d$^{-1}$, Eq.17). Firstly, the fast groundwater recharge $R_{fs}$ (mm d$^{-1}$, Eq. 15) is defined by the division parameter (1-$C_p$). The remaining water $R_{ff}$ (mm d$^{-1}$, Eq. 15) is then further divided to account for infiltration-excess overland flow $Q_o$ (mm d$^{-1}$, Eq. 16) which is defined by the division parameter ($C_n$) and the threshold parameter $P_{tresh}$ (mm d$^{-1}$) (Horton, 1933). We assumed a constant value for the division parameter ($C_n$) to limit the number of calibration parameters in the spirit of model

parsimony. After subtraction of fast groundwater recharge and overland flow, the remaining fast and lateral subsurface flux $R_{fn}$ (mm d$^{-1}$,Eq.17) enters the fast storage component $S_f$ (mm, Eq. 4). If the maximum capacity of $S_f$ (mm, Eq. 4) is exceeded, water is released as saturation excess overland flow $Q_{of}$ (mm d$^{-1}$, Eq. 18). Otherwise, it is released to the stream as fast flow $Q_f$ (mm d$^{-1}$, Eq. 19).

Groundwater storage was separated into an "active" groundwater storage $S_{s,a}$ and a hydrologically "passive"

storage volume $S_{s,p}$ (mm). $S_{s,p}$ (mm) does not change over time if there are no deep infiltration losses, so that $dS_{S,p}/dt=0$ (Zuber, 1986; Hrachowitz et al., 2016). This "passive" storage does not contribute to runoff but its role is to isotopically mix water of the "active" storage with water of the "passive" storage which is represented as $S_{s,tot} = S_{s,a} + S_{s,p}$. The use of the total groundwater storage $S_{s,tot}$ facilitates contributions from both $S_{s,a}$ and $S_{s,p}$ to the age structure of the outflow $Q_s$ (mm d$^{-1}$, Eq. 24). Water enters the groundwater storage as sum of slow





percolation $R_s$ (mm d$^{-1}$) and fast recharge $R_{fs}$ and is released as base flow $Q_s$ (mm d$^{−1}$, Eq. 24) and deep infiltration losses $Q_l$ (mm d$^{-1}$, Eq. 25).

### 2.3 Tracer transport model

#### 2.3.1 Rank StorAge Selection (rSAS) function

We combined the hydrological model as described in the previous chapter with a transport model that utilizes
the age-rank StorAge Selection (rSAS) function which ranks stored water volumes by age (Harman, 2015; Benettin et al., 2017) to capture the variability of outflow age over time. The general theoretical framework of the transport model relies on the studies of Botter et al. (2009), van der Velde et al. (2012), Harman (2015) and Benettin et al. (2015). At any given time t, each storage $S_{T,m,j}(t)$ defined within the hydrological model (Fig. 2) stores water of different ages. That is represented as T and traces back to past precipitation inputs at age T = 0.
The age distribution of storage at time t is termed $p_s(T,t)$. The outfluxes (e.g., evapotranspiration and discharge) consist of specific age subsets from the storage, resulting in distinct age-ranked distributions for the water leaving the storage. These are termed $p_{E,T}(T,t)$ for evapotranspiration and $p_{Q,T}(T,t)$ for discharge. At each given time t, the total water volume in storage is also characterized by its tracer composition and distributions $C_S(T,t)$ which traces back to past precipitation inputs. In the case of an ideal tracer, it is equal to the water stable isotope
composition of past precipitation ($P_{\delta^{18}O}$) upon entering the catchment at time t-T, i.e., $C_P(t−T)$. As a result, output fluxes are characterized by water stable isotope compositions ($Q_{\delta^{18}O}$, $Q_{\delta^2H}$) which is $C_Q(t−T)$ for streamflow and ($ET_{\delta^{18}O}$, $ET_{\delta^2H}$) which is $C_{ET}(t−T)$ for evapotranspiration.

#### 2.3.2 Integration of rank StorAge Selection (rSAS) function concept and hydrological model

The water age balance (Equation 2727) is formulated individually for each of the j storage components of the
model such as canopy interception or the root zone, based on their transport dynamics. The change in water storage is the difference between age-ranked input volumes $I_{T,j}(T, t)$ (mm d$^{-1}$) and age-ranked output volumes $O_{T,j}(T, t)$ (mm d$^{-1}$) (Botter et al., 2011; Harman, 2015; and van der Velde et al., 2012).

$$\frac{\delta S_{T,j}(T,t)}{\delta t} + \frac{\delta S_{T,j}(T,t)}{\delta T} = \sum_{n=1}^{N} I_{T,n,j}(T,t) - \sum_{m=1}^{M} O_{T,m,j}(T,t)$$
(27)

$\partial S_{T,j}(T,t)/\partial T$ is the aging process of water in storage, N and M are number of inflows and outflows from that
storage component (e.g., for the root zone these would be $E_a$, $R_f$, and $R_s$ (Fig. 3). Each age-ranked outflow $O_{T,m,j}(T, t)$ (Equation 28) from a specific storage component j (Fig. 3) depends on the outflow volume $O_{m,j}(t)$ which is estimated by the hydrological balance component of the model (see chapter 2.2) and the cumulative age distribution $P_{o,m,j}(T, t)$ of that outflow.

$$O_{T,m,j}(T,t) = O_{m,j}(t)P_{O,m,j}(T,t)$$
(28)

The cumulative age distribution $P_{o,m,j}(T, t)$ (Equation 29), which is the backward transit time distribution TTD of that outflow in cumulative form, depends on the age-ranked distribution of water in the storage component j, represented by $S_{T,j}(T, t)$ for time step t and probability density function, which in this case is SAS function $\omega_{o,m,j}$ (or $\Omega_{o,m,j}$ in its cumulative form) of that flux.




$$P_{O,m,j}(T,t) = \Omega_{O,m,j}(S_{T,j}(T,t),t) \tag{29}$$

The SAS function $\omega_{o,m,j}$ (or $\Omega_{o,m,j}$ in its cumulative form) is a probability density function of normalized rank storage $S_{T,\text{norm,j}}(T,t)$ (Equation 31) at time t, which can also be formulated as residence time distribution RTD of storage component j (e.g., root zone) at time t (Equation 30). Normalizing the age-ranked storage helps prevent rescaling $\omega_{o,m,j}$ at each time step to conserve mass balance. Therefore, we used normalized rank storage (Equation 5) to bind the age-ranked storage to the interval [0,1].

$$p_{O,m,j}(T,t) = \overline{\omega}_{O,m,j}(S_{T,j}(T,t),t)\frac{\delta S_{T,j}}{\delta T} \tag{30}$$

$$S_{T,norm,j}(T,t) = \frac{S_{T,j}(T,t)}{S_j(t)} \tag{31}$$

$\delta^{18}O$ signals from entering the catchment as precipitation to leaving it as streamflow can be tracked through each
individual storage component based on the tracer balance (Equation 32)(e.g., Harman, 2015; Benettin et al., 2017).

$$C_{O,m,j}(t) = \int_0^{S_j} C_{S,j}(S_{T,j}(T,t))\omega_{O,m,j}(S_{T,j}(T,t),t)dS_T. \tag{32}$$

Where $C_{o,m,j}$ is the $\delta^{18}O$ composition in outflow m from storage component j at time t, $C_{s,j}$ is the $\delta^{18}O$ composition of water in storage at time t

### 2.3.3 Time-variable and conditional SAS functions

Previous studies found a difference in transport processes between wet and dry periods (Weiler and McDonnell, 2007; Beven, 2010; Beven and Germann, 2013; Klaus et al., 2013; Loritz et al., 2017; Hrachowitz et al., 2021).
This suggests that SAS functions are also time-variable and can be formulated as varying between preferential release of younger water, preferential release of older water or no preference (uniformly selected) (van der Velde et al., 2012; van der Velde et al., 2015; Hrachowitz et al., 2016). In this study, we used a beta distributions with shape and scale parameters α ( − ) and β ( − ) as SAS functions. When both parameters of beta distributions are equal to 1 (α = b = 1), this indicates no selection preference for specific ages (uniform
selection). If α < b (or α > β), it indicates a selection preference for younger (or older) water. To limit the number of parameters, we kept "b" equal to 1. The time variability of the SAS function shape is then based on age rankStorage and the shape parameter (α) which is bounded [0,1] for the preference of younger storage and bounded [α>1] for the preference of older storage. In the following we use this approach for the root zone storage Sr.



In contrast, all other storage components (e.g., snow, groundwater) were based on uniform sampling ($\alpha = 1$, $\beta = 1$). Despite the shape parameters being fixed to uniform sampling in each of these storage components, the resulting overall SAS function, aggregating the individual storage components, is nevertheless time-variable due to the different time scales of and the temporally varying contributions from the individual components (Equation 30).

Previous studies have shown that as soil moisture increases, preferential flow increasingly bypasses small pore volumes, leading to the release of younger water (Weiler and McDonnell, 2007; Beven, 2010; Loritz et al., 2017; Hrachowitz et al., 2021). To mimic this behaviour, SAS functions for the fast preferential flow $R_f$ (mm d$^{-1}$), were formulated with a time-variable shape factor $\alpha$ (t) (Fig. 4), which varied between 0 to 1 for each time step t. The variation of $\alpha(t)$ was done by following Hrachowitz et al. (2013) and van der Velde et al. (2015), by 310 varying it as a function of the stored water volume $S_r$ (t) and the maximum storage capacity ($S_{r,max}$) as shown in Equation 33 and Figure 4 (Scenario 1):

$$\alpha(t) = 1 - \frac{S_r(t)}{S_{r,max}}(1 - \alpha_0)$$

(33)

where $\alpha_0$ is a calibration parameter representing a lower bound between [0,1], so that $\alpha(t)$ can vary between $\alpha_0$ and 1; $\alpha(t) = 1$ indicates a uniform sampling SAS function at low soil moisture (dry soil) (Fig. 4a, A). This 315 formulation (Scenario 1, Figure 4a) leads to an increasing preferential release of younger water as the system becomes wetter.

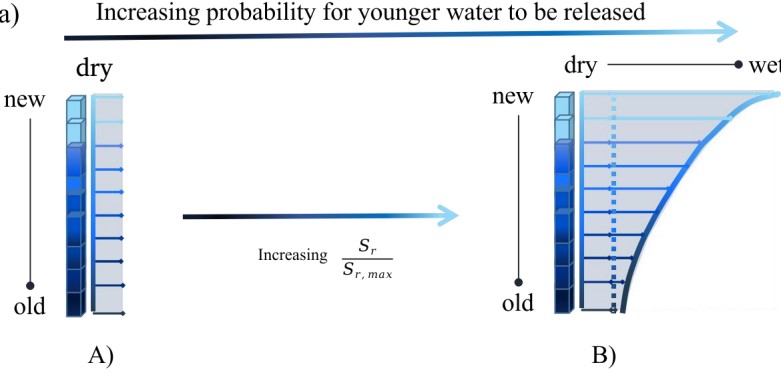

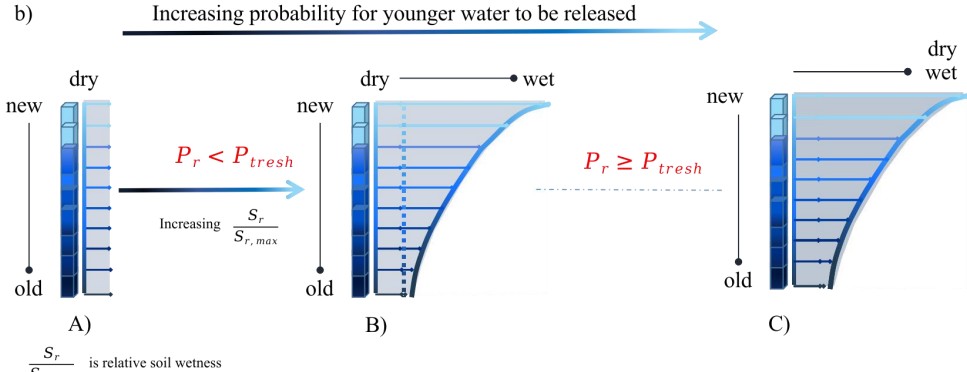

**Figure 4.** The two tested scenarios for determining the shape of the time-variable SAS function for fast flux $R_f$ (mm
d$^{-1}$) (Fig. 3). The age-ranked storage probability function is shown as vertical bars in all panels (A,B,C), with the
light blue color representing young water (at the top of the vertical bars), while the dark blue color represents old
water (at the bottom of the vertical bars). (a) Scenario 1 (S1), the time-variable SAS function depends on the ratio of
current storage $S_r$ to maximum storage capacity $S_{r,max}$ with the preference for young water increasing as storage
increases from A to B and (Equation 33). (b) Scenario 2 (S2), the condition (A to B) only applies only when
precipitation intensity does not exceed the threshold intensity ($P_{tresh}$). If precipitation intensity exceeds the threshold
intensity ($P_{tresh}$), young water is preferred with higher probability (C) regardless of the current wetness state. This
mimics the rainfall bypassing the soil storage as fast overland or subsurface lateral flow.

Previous research highlighted the non-linearity of flow processes in the HOAL catchment, where precipitation
can quickly generate fast runoff and bypass the soil storage as fast overland or subsurface lateral flow (Blöschl
et al., 2016; Exner-Kittridge et al., 2016; Vreugdenhil et al., 2022; Hövel et al., 2023; Szeles et al., 2024). To
mimic and test this in our study, SAS functions for the fast preferential flow Rf (mm d$^{-1}$), were formulated with
a time-variable shape factor α (t) vary as a function of soil moisture as it is in Equation 33 (Scenario 1, Figure
4a), but additionally became equal to α0 (-) (lower bound) when precipitation intensity PI (mm d-1) exceeded a
certain threshold Ptresh (Scenario 2, Figure 4b, Equation 34) .

$$\alpha(t) = \begin{cases} \alpha_0, & \text{if } P_r(t) \geq P_{tresh} \\ 1 - \frac{S_r(t)}{S_{r,max}}(1-\alpha_0), & \text{if } P_r(t) < P_{tresh} \end{cases} \tag{34}$$

This formulation (Scenario 2, Figure 4) leads to an increasing preferential release of younger water with
increasing soil moisture. Additionally, higher probability of release of younger water bypass the soil stored
water when precipitation intensity $P_I$ (mm d$^{-1}$) exceed the threshold intensity ($P_{tresh}$). This formulation mimic
rainfall bypassing the soil storage as fast overland or subsurface lateral flow.

## 2.4 Model optimization

The model was run with a daily time step for the time period between October 2013 and 30 December 2018 to
calibrate the 15 hydrological and 2 tracer transport parameters model parameters (Table 2). We used the 1 year
data from October 2013 to October 2014 as warm-up period. Using an objective criteria that combines 6



performance criteria (Table 3) related to streamflow and tracer dynamics, we implemented the Differential
Evolution algorithm (Storn and Price, 1997) to optimize model parameters. For model calibration and evaluation,
we used six performance metrics (Table 3) that describe the model's ability to simultaneously reproduce
different signatures associated with streamflow Q (mmd$^{-1}$) and $\delta^{18}O$ dynamics of the streamflow (Eq. 35). These
are the Nash-Sutcliffe Efficiencies (NSE) (Nash and Sutcliffe, 1970) of streamflow, of the logarithmic
streamflow, of the flow duration curve and of the time series of seasonal runoff ratios (averaged over three
months). For $\delta^{18}O$ signals we used the Nash-Sutcliffe efficiency (NSE) of $\delta^{18}O$ all measured samples (daily
event and weekly grab samples) (Fig. 2c) and the mean square error of weekly grab samples (Fig. 2d). The
individual performance metrics were aggregated into the Euclidean Distance DE to the perfect model, using
equal weights for the 6 stream flow and 2 tracer signatures, respectively, according to:

$$DE = \sqrt{\frac{1}{2}\left(\sum_{i=m}^{M}\frac{(1-E_{Q,m})^2}{M} + \sum_{i=n}^{N}\frac{(1-E_{18O,n})^2}{N}\right)}$$

(35)

where M is the number of performance metrics with respect to streamflow, N is the number of performance
metrics for tracers in each combination, and E is the evaluation matrix based on goodness of fit criteria. DE is
Euclidean distance to the 'perfect model', with zero indicating a perfect fit. We selected the 50 best parameter
sets ranked by decreasing Euclidean distance DE for model evaluation.

We used two scenarios for model calibration where the formulation for hydrological fluxes were identical but
transport formulation were different for SAS function shape lower bound $\alpha_0$ (-) as described in section 2.3.3:
Scenario 1 (S1), with $\alpha(t)$ as a linear function of wetness ($S_r/S_{r,max}$) (Equation 33), and Scenario 2 (S2), with $\alpha(t)$
being a linear function of wetness ($S_r/S_{r,max}$) if precipitation intensity is less than threshold intensity ($P_{tresh}$).
However if precipitation intensity exceed the threshold intensity ($P_{tresh}$) $\alpha(t)$ was formulated as strong preference
for young water with shape factor $\alpha(t)= \alpha_0$ (-) (Equation 34).

**2.5 Model comparison and data analysis**

We evaluated the performance of the model under two scenarios using six performance metrics, which are listed
in Table 3 for the tracking period from October 2014 to December 2018. Next, we analyzed transit times in
relation to hydrological and hydroclimatic drivers by categorizing water into different age thresholds. These
thresholds included: T<7 days, representing "event" water; 7<T<90 days, representing young water with some
delay; and 90<T<365 days, representing longer transit times. The streamflow age fraction $F_Q$ (T<$T_{age}$ days) is
calculated based on the sum of TTD, where T<$T_{age}$ days. For example, the age fraction of streamflow $F_Q$ (T<90
days) is calculated based on the sum of TTDs, where T<90 days. We calculated the mean and maximum
percentage of streamflow fractions for transit times T<7 days, T<90 days, 7<T<90 days, and 90<T<365 days.
We also compared the variation in mean and maximum percentage of streamflow water age fractions for
different seasons autumn (September, October, November), winter (December, January, February), spring
(March, April, May), and summer (June, July, August) as well as for distinct wetness states (dry, drying, wet
wetting periods). Dry days were marked by flows less than the 25th quantile, while wet days were marked by





flows higher than the 75th quantile. Drying days marked any decay between the 25th quartile and 75th quartile whereas wetting days are marked as any increase between the 25th quartile and 75th quartile.

Furthermore, we compared the relationship between transit times and hydrological and hydroclimatic drivers, specifically, streamflow Q (mm d$^{-1}$), precipitation intensity (mm d$^{-1}$), and volumetric soil water content SWC (%) for the tracking period as well as across different seasons and wetness states to understand variations in the control mechanisms. This analysis was conducted by comparing Spearman rank correlation coefficients of water age fractions with the hydroclimatic drivers.

**Table 3:** Signatures for streamflow, $\delta^{18}$O signal and the associated performance metrics used for model calibration scenarios and evaluation.

| Signatures | Abbreviation | Performance Metric | Reference |
|---|---|---|---|
| Time series of streamflow | Q | $NSE_Q$ <br> $NSE_{(logQ)}$ | Nash and Sutcliffe (1970) |
| Flow duration curve | FDC | $NSE_{FDC}$ | Jothityangkoon et al. (2001) |
| Seasonal runoff ratio | RC | $NSE_{RC}$ | Yadav et al. (2007) |
| Times series $\delta^{18}$O in streamflow | $\delta^{18}$O | $NSE_{\delta^{18}O}$ <br> $MSE_{\delta^{18}O}$ | Birkel et al. (2011a) |

## 3 Results

### 3.1 Model calibration results

The model parameters selected for the HOAL catchment for calibration period from October 2014 to December 2018 reproduced the general features of the hydrograph (Fig. 5). The best-performing model generally captured both the timing and magnitude of high and low flow events independent on the selected scenario ($NSE_Q = 0.61$ for both scenarios, Figure 5a), with the exception of over-estimations of low flows during the summer 2016 and underestimation of low flows during the winter 2017. The three-month averaged runoff ratio (RC) was

reproduced, with NSE values of 0.89 for Scenario 1 and 0.83 for Scenario 2 (Fig. 5b, e). The flow duration curve (FDC) was reproduced, with Nash-Sutcliffe efficiency ($NSE_{FDC}$) of 0.51 for Scenario 1 and 0.50 for Scenario 2 (Fig. 5d,e). Low flows were reproduced, with a median Nash-Sutcliffe efficiency of log-flows ($NSE_{logQ}$) as 0.65 (Fig. 5c,e). For several rain storms, the model reproduced the sharp $\delta^{18}$O fluctuations during events and a highly stable $\delta^{18}$O signal between consecutive events (Fig. 5c) for both scenarios. However, S2

indeed showed considerable improvements for the very negative winter $\delta^{18}$O stream values in 2015 and 2018, but also for several events in summer 2016, 2017 and 2018. The performance metrics based on median $\delta^{18}$O





signals were higher for Scenario 2 with e.g. $NSE_{\delta^{18}O} = 0.51$ than for Scenario 1 with $NSE_{\delta^{18}O} = 0.31$ (Fig. 5e). Overall, the Euclidian distance DE for 50 best performing parameter sets decreased from 0.42 for Scenario 1 to 0.37 for Scenario 2, showing that Scenario 2 performed generally better than Scenario 1 (Fig. 5e).


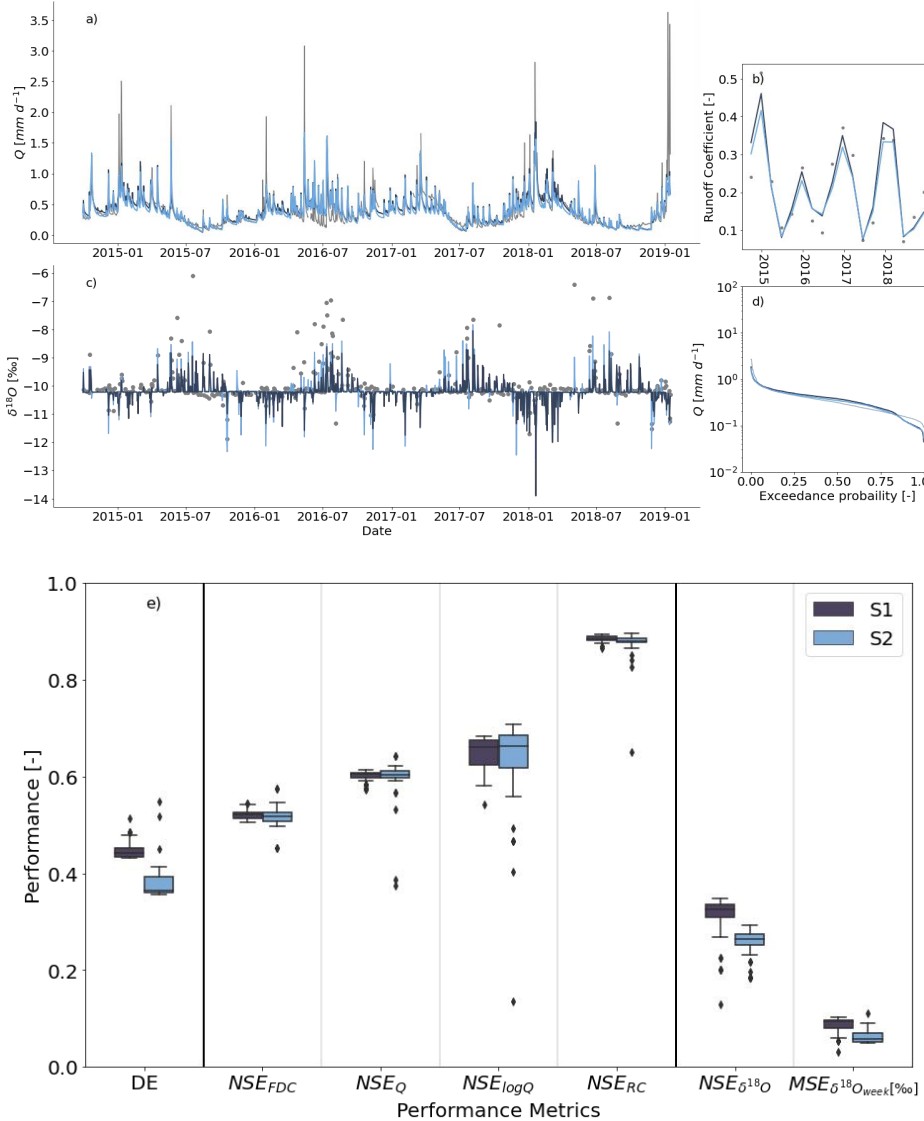

**Figure 5.** Model calibration results for Scenario 1 (S1, dark blue) and Scenario 2 (S2, light blue) (a, d) where the observed values are shown as gray dots and lines. (a) streamflow [mm d$^{-1}$], (b) streamflow $^{18}$O [‰], (c) the three-month average runoff coefficient RC [-], (d) the flow duration curve [mm d$^{-1}$], and (e) boxplots of performance metrics of the two scenarios based on 50 best performing parameter sets.






### 3.2 Water transit times and residence times

By tracking the $\delta^{18}O$ signals through the model, we estimated TTDs in streamflow and compare these
distributions for different age thresholds, T< 7 days, 7<T<90 days, T<90 days, and 90<T<365 days (see Section
2.4). It is important to acknowledge that the transit time results are inherently tied to the assumptions made and
the uncertainties within the modeling process. Model calibration based on Scenario 2 resulted in more younger
water bypassing storage as evidenced by the mean percentage of streamflow age fraction younger than 7 days
$F_Q$(T<7days) being lower for Scenario 1 (2.87%) compared to Scenario 2 (4.03%) (see Table 4 and Figure, 6a,
Figure, S2). This is also reflected in individual TTDs for fast preferential flow Rf (mmd $^{-1}$)(Fig. 3), where on
average 40% of fast preferential flow was from recent rainfall (age = 1 day) based on S2 and was 30% for S1
(Fig. 6e). However, the scenarios did not differ in the fraction of streamflow that is younger than 90 days
$F_Q$(T<90 days) where the mean percentage for Scenario 1 was 6.03% and for Scenario 6.53 % (see Table 4 and
Figure 6b).

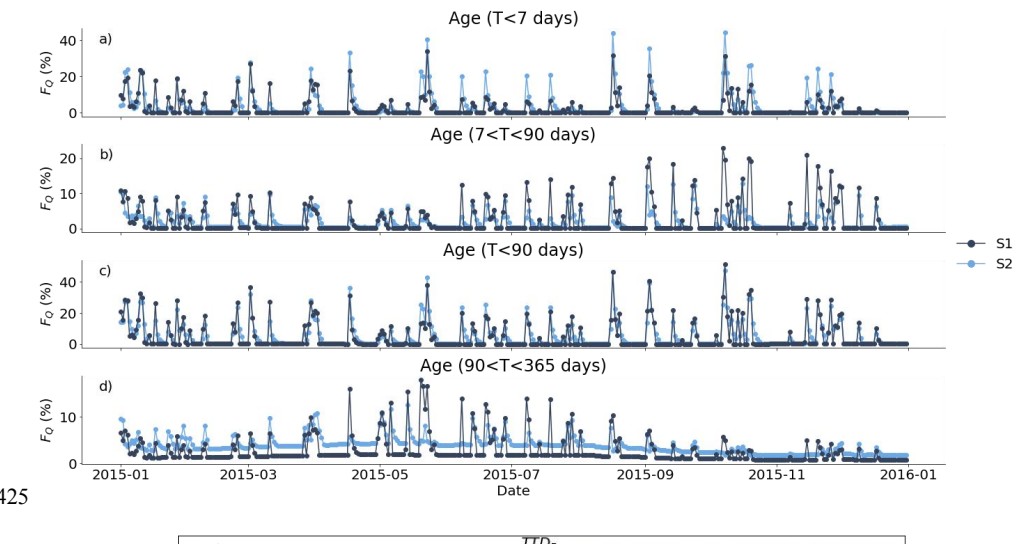


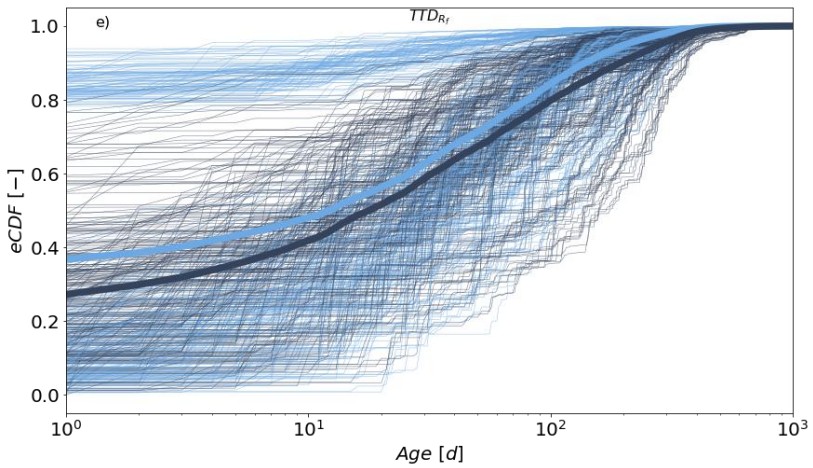





**Figure 6.** The percentage of water age fractions based on two scenarios for the year 2015 (a, d). The result for the full calibration period can be found in Supplementary Figure S2. (a,e), dark blue dots represent the results from Scenario 1 (S1) and light blue dots represent the results from Scenario 2 (S2). The age fraction of streamflow are categorized by age:(a) T< 7 days, (b) 7<T<90 days, (c) T<90 days, and (d) 90<T<365 days. Panel (e) shows individual transit time distributions (TTD) based on Scenario 1 and Scenario 2 for total fast recharge $R_f$ (Fig. 3) as cumulative distribution functions eCDF(-). The bold lines in panel (e) are mean of individual TTDs in cumulative form based on based on Scenario 1 (dark blue line) and Scenario 2 (light blue line).

**Table 4:** Summary of the mean and maximum (max) percentage of water transit times (categorized by T<90, 0< T<7, 7<T<90, 90<T<365 in days) based on Scenario 1 and Scenario 2.

| Transit time (day) | S1 | | S2 | |
|---|---|---|---|---|
| | mean (%) | max (%) | mean (%) | max (%) |
| T<90 | 6.03 | 52.99 | 6.53 | 48.47 |
| 0<T<7 | 2.87 | 36.41 | 4.38 | 45.89 |
| 7<T<90 | 2.83 | 25.73 | 2.15 | 14.46 |
| 90<T<365 | 2.67 | 24.90 | 3.59 | 17.27 |

### 3.3 Influence of hydrological and hydroclimatic variables on water age fractions

The influence of hydrological and hydroclimatic variables on water age fractions (0< T<7, T<90, 90<T<365 in days) were compared by Spearman rank correlation coefficients (r, p). Only precipitation intensity $P_I$ (mm $d^{-1}$) was strongly correlated with the streamflow water age fraction younger than 7 days $F_Q$ (T<7days) for both scenarios, with a slightly higher correlation coefficients for Scenario 1 (S1, r = 0,67 p < 0.05) compared to Scenario 2 (S2, r = 0.53, p <0.05) (Fig. 7b). Similarly, water age fractions younger than 90 days $F_Q$ (T< 90 days) were more correlated with precipitation intensity $P_I$ (mm $d^{-1}$) than with volumetric soil water content SWC (%) or streamflow Q (mm $d^{-1}$) (Fig. 7d, 7e, 7f). The correlation coefficients (r) with precipitation intensity $P_I$ (mm $d^{-1}$) were r = 0.71, p < 0.05 for Scenario 1 as and were r =0.62 , p < 0.05 for Scenario 2. For streamflow age fractions between 90 and 365 days $F_Q$ (90<T<365 days) only Scenario1 resulted in strong correlation coefficients with precipitation intensity $P_I$ (mm $d^{-1}$) (r = 0.58, p < 0.05) (Fig. 7h). No strong correlations were found for all other combinations of the water age fractions to streamflow Q (mm $d^{-1}$) or volumetric soil water content SWC (%) (Fig. 7).

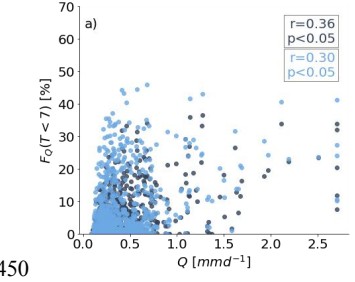
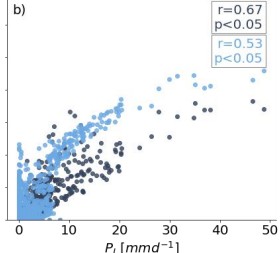
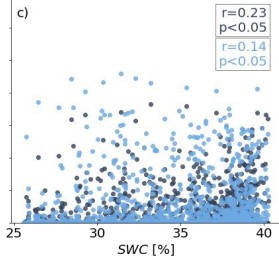



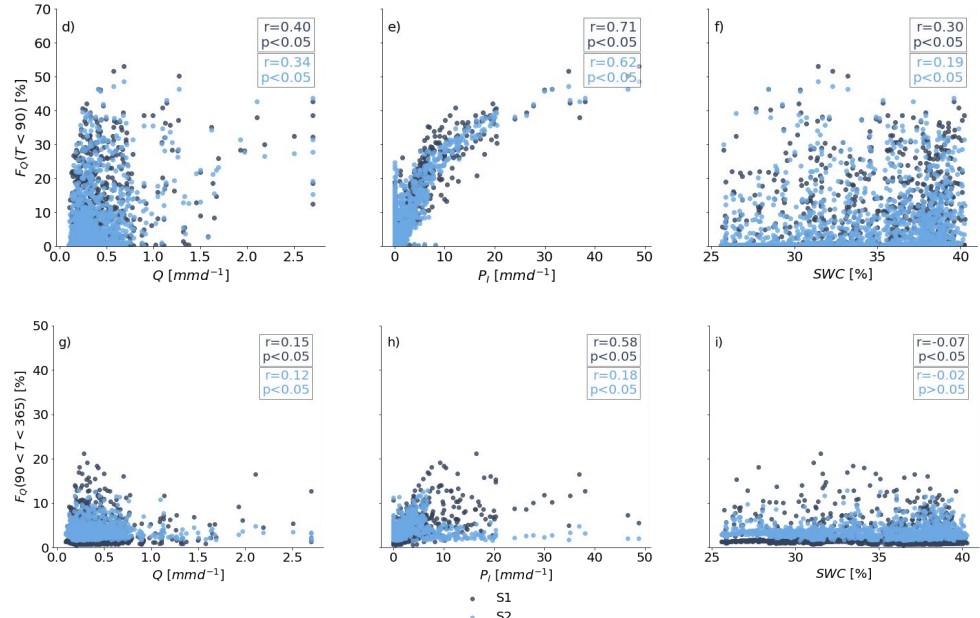

**Figure 7.** Spearman rank correlation of streamflow water age fractions with the hydrological and hydroclimatic variables, discharge Q [mmd$^{-1}$], precipitation intensity $P_I$ [mmd$^{-1}$], and volumetric soil water content SWC [%]. Panel (a, b, c) show the correlations of streamflow age fractions younger than 7 days $F_Q$ (T<7days), (d, e, f) show the correlations of streamflow age fractions younger than 90 days $F_Q$ (T< 90 days) and (g, h, i) correlations of streamflow age fractions older than 90 days but younger than 365 days $F_Q$(90<T<365 days) to discharge Q [mmd$^{-1}$], precipitation intensity $P_I$ [mmd$^{-1}$], and volumetric water content SWC [%] respectively.

### 3.4 Linking water age fractions to hydrological and hydroclimatic drivers in different seasons

Scenario 2 resulted in a higher fraction of streamflow water younger than 7 days $F_Q$ (T<7days) , especially during autumn and summer, compared to Scenario 1 (Fig. 8a, 9a). However, during spring and winter, both scenarios reproduced similar results for $F_Q$ (T<7days). On average, ~2% and ~4% of autumn recharge was younger than 7 days based on Scenario 1 and Scenario 2 respectively. For individual events, these values reached up to a maximum of 31% and 44 % based on Scenario 1 and Scenario 2 respectively (Table 5).

Similarly, in the summer season, Scenario 2 resulted in a higher fraction of streamflow younger than 7 days with average of 4.48% compared to Scenario 1 (~3%). For water ages 7<T<90 days and 90<T<365 days, Scenario 1 resulted in higher fractions across all seasons compared to Scenario 2 (Fig. 9b, 10c; Table 5)

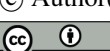

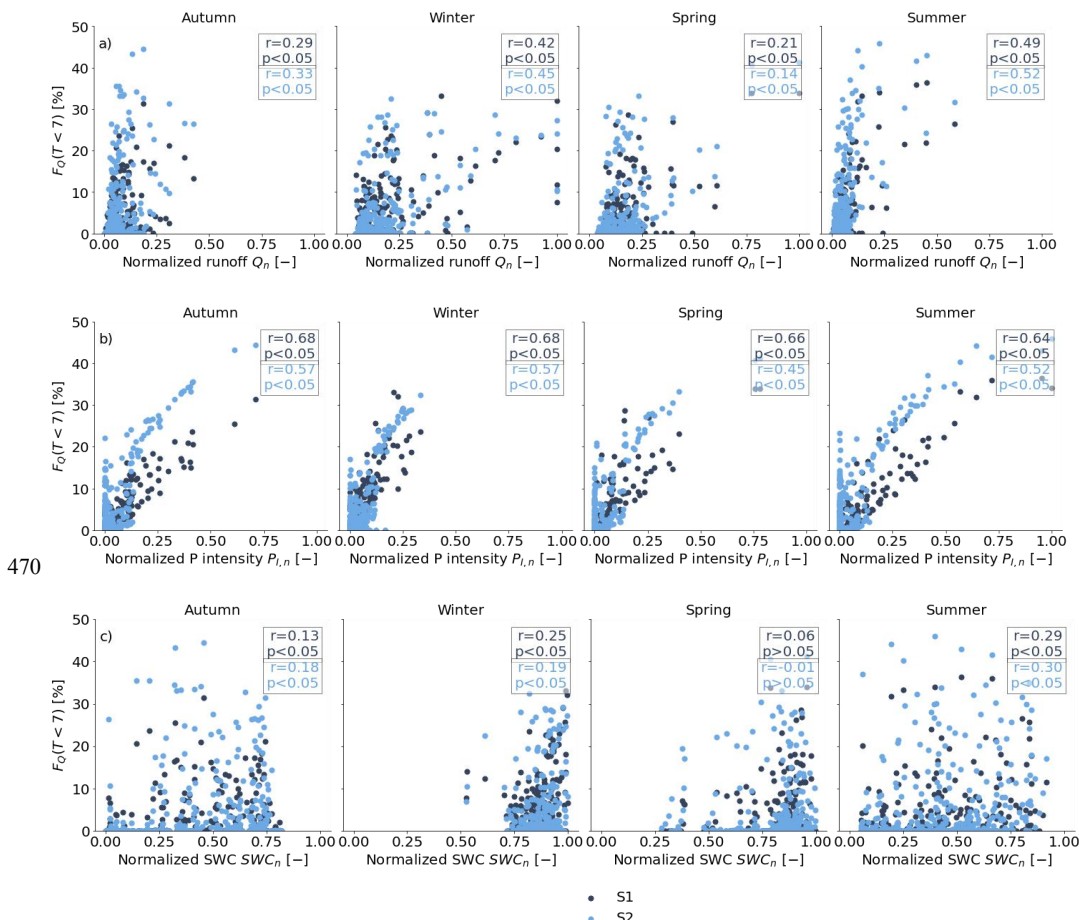


**Figure 8.** Spearman rank correlation of streamflow water age fractions younger than $F_Q$ (T<7 days) with hydrological and hydroclimatic variables across different seasons (Autumn, Winter, Spring, Summer). (a) Normalized discharge Q [-]) correlations to $F_Q$ (T<7 days) in different seasons, (b) normalized precipitation intensity $P_{I,n}$ [-] correlations to $F_Q$ (T<7 days)

in different seasons (c), and normalized volumetric water content SWC [-] correlations to $F_Q$ (T<7 days) in different season.

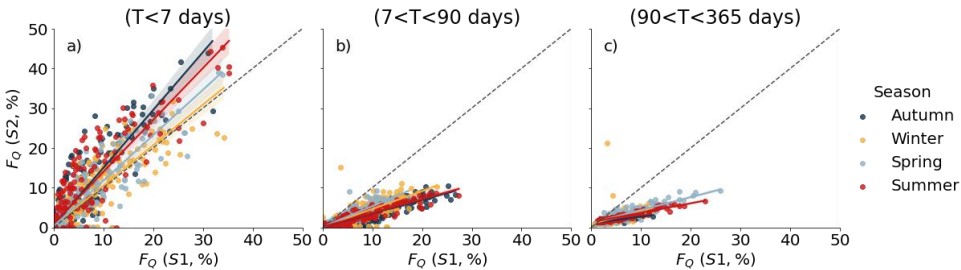





**Figure 9:** Comparison of estimated water ages based on two scenarios. Streamflow age fraction results from Scenario 1 are represented on the x-axis, while results from Scenario 2 are represented on the y-axis. The black dashed lines represents the 480 1:1 line for all panels. The comparison of estimated water age fractions younger than 7 days (a), age fractions from 7 to 90 days (b), and age fractions between 90 to 365 days (c).The colors indicate different seasons (dark blue: Autumn, yellow: Winter, light blue: Spring, and red: Summer)

**Table 5:** Summary of the mean and maximum (max) percentage of water transit times (categorized by age 0< T<7, 7<T<90, 90<T<365 in days) based on Scenario 1 and Scenario 2 for autumn, winter, spring and summer.

| Transit time (day) | | S1 | | | | S2 | | | |
|---|---|---|---|---|---|---|---|---|---|
| | | Autumn | Winter | Spring | Summer | Autumn | Winter | Spring | Summer |
| 0<T<7 | mean(%) | 2 | 4 | 3 | 3 | 4 | 4 | 4 | 5 |
| | max (%) | 31 | 33 | 34 | 36 | 44 | 32 | 41 | 46 |
| 7<T<90 | mean(%) | 3 | 3 | 2 | 3 | 2 | 3 | 2 | 2 |
| | max (%) | 26 | 20 | 15 | 24 | 14 | 13 | 10 | 13 |
| 90<T<365 | mean(%) | 2 | 2 | 3 | 3 | 3 | 4 | 4 | 4 |
| | max (%) | 12 | 11 | 25 | 21 | 7 | 17 | 15 | 13 |


## 4 Discussion

### 4.1 Soil moisture is not the only control of transit times

Previous studies have shown that soil moisture plays a significant role in catchment transit times in humid areas such as Wüstebach and the Bruntland Burn catchment in Scotland (Benettin et al., 2017; Hrachowitz et al., 490 2021). However, in the HOAL catchment of this study, rainfall intensity, beyond soil moisture, was required to account for the complexity of the hydrological and transport response.

For both scenarios (S1: SAS function with soil moisture only, S2: SAS function with soil moisture and rainfall intensity), the mean fraction relatively short travel times in stream water (T<7, 7<T<90, and T<90 days) positively correlated with modeled soil moisture (Table S1). This suggests that catchment soil moisture plays a 495 role for young water release in the HOAL catchment, which is further supported by the reasonably good simulation results of stable isotopes of water when only using soil moisture in the SAS function (NSE = 0.31). Therefore, the results correspond well to earlier research, where increasing catchment wetness resulted in younger water reaching the stream (Weiler and Naef, 2003; Zehe et al., 2006; Hrachowitz et al ., 2013; Remondi et al., 2018; Rodriguez et al., 2018; Sprenger et al., 2019).

Despite the selection of the SAS function based exclusively on catchment wetness being adequate for the HOAL catchment, the highly complex runoff generation mechanisms (Blöschl et al., 2016) with a quick runoff response particularly during autumn and summer months, highlighted the need for an additional control on the SAS function shape (Fig. 5c, 6a). Indeed, the model performance was better (Figure 5e) when including precipitation intensities in the SAS function (Figure 4b ). This indicates that the direct contribution of precipitation to 505 streamflow during storm events with high precipitation intensities is important in the HOAL catchment. This behavior can be explained by several factors that promote fast runoff that bypass resident water.





The incorporation of both soil moisture and precipitation intensity in the SAS function accounts for non-linearity of flow processes, mimicking the behaviour of not only saturation-excess overland flow but also that of infiltration-excess flow and other subsurface fast runoff flow processes that bypass flow with minimal interaction with resident water (e.g. tile drain flow). Therefore, we included a non-linear threshold behavior in the SAS function with rainfall intensity, where changes in runoff processes or shifts in runoff regimes can occur. The non-linearity of flow processes in the HOAL catchment has been demonstrated through hydrometric analysis and visual observations, which have highlighted the potential controls of soil moisture and event precipitation (Blöschl et al., 2016; Exner-Kittridge et al., 2016; Vreugdenhil et al., 2022; Hovel et al., 2023; Szeles et al., 2024). Similarly, Vreugdenhil et al. (2022) showed that rainfall and soil moisture are significant and highly non-linear controls on overland flow and tile drainage flow in different parts of the HOAL used here. For instance, tile drainage in wetlands was more linearly related to soil moisture, whereas at the hillslope scale, it was more related to precipitation even at low-intensity rainfall. Therefore it is plausible to assume that in the HOAL catchment overland flow exhibits a threshold behavior related to fast runoff generation occurring even at low-intensity rainfall.

Additionally, the HOAL catchment consists of a diverse range of soil types, with a high clay content between 20% and 30% (Blöschl et al., 2016). Different types of soils may introduce complexities due to surface and subsurface heterogeneity in soil hydraulic conductivity, which significantly influences the shapes of SAS functions (Danesh-Yazdi et al., 2018). As previously discussed by Danesh-Yazdi et al. (2018), subsurface heterogeneity in hydraulic conductivity imposes significant variation in the shape of the SAS function. Therefore, assuming a smooth functional form for the SAS function in heterogeneous systems may oversimplify its intrinsic variability concerning age or age-ranked storage. This may also explain why incorporating soil moisture and precipitation intensity, as we did in Scenario 2, resulted in better model performance in the simulation of the $\delta^{18}O$ signal in streamflow.

Besides, the tile drainage system, which covers only 15% of the catchment (Fig.1), appeared to play an important role in fast flow generation. The close resemblance of the $\delta^{18}O$ signal in the tile drainage system with the precipitation $\delta^{18}O$ signal (Fig. 2e, 2f) provides evidence that some event precipitation contributes to the stream through the tile drain not only in winter but also in summer. A possible explanation for summer months is that larger cracks in the clayey soils, which are directly connected to the tile drainage system, allow for preferential flow that is more dependent on precipitation intensity than on soil moisture. This results corresponds with observations from Exner-Kittridge et al. (2016), who noted that in the HOAL catchment, macropore flow is observed in summer when the topsoil dries and forms cracks due to high clay content. This emphasizes the critical role of soil texture and structure in influencing water movement during rainfall events.

## 4.2 Synthesis of streamflow generation processes in the HOAL catchment

The HOAL catchment exhibits a diverse and rapid hydrological response to precipitation events (Blöschl et al., 2016; Exner-Kittridge et al., 2016; Vreugdenhil et al., 2022). This is also evidenced by the on/off response of streamflow and the sharp transition between high-resolution event $\delta^{18}O$ signals and highly stable weekly $\delta^{18}O$ signals observed in the stream (Figure 2c, 2e, 2f). Tracer compositions measured at weekly intervals remained stable stable throughout the year (Fig. 2d). However, event-based samples and tile drainage samples showed





similar $\delta^{18}$O patterns to precipitation (Fig. 2f), indicating a sharp transition between fast flow processes and more stable groundwater flow. For several rain storms, the model reproduced the sharp fluctuations during events and a stable $\delta^{18}$O signal between consecutive events (Fig. 5c) for both scenarios. Nevertheless, the model calibration based on Scenario 2 enhanced the model's sensitivity to the time scale of fast flow (Fig. 6a), further emphasizing the critical role of precipitation intensity in influencing hydrological responses in the catchment. In
particular, infiltration-excess overland flow and precipitation-driven subsurface fast flow were identified as key flow processes, corroborating studies by Blöschl et al. (2016), Széles et al. (2020), and Silasari et al. (2017), who noted that both saturation-excess and infiltration-excess overland flow typically occur in valley bottoms during prolonged or intensive rainfall, with part of the event water entering the stream as overland flow. The hydrological behavior of the HOAL catchment supports earlier findings by Kirchner et al. (2023), who noted
that a rapid hydrological response often indicates rainwater quickly moving to channels via overland flow or fast subsurface pathways.

### 4.3 Catchment transit times

Transit time results indicated that event peaks were primarily a mixture of new precipitation water and the water less than 7 days old that had been stored in the catchment. During events, the percentage of streamflow water
age fractions for T<7, 7<T<90, and 90<T<365 days increased for both scenarios (Fig. 6a). However, on average, only ~ 4% of the water was younger than 7 days, and ~7% was younger than 90 days (Table 4). This aligns with the findings of previous studies that have identified the majority of water contributing to streamflow as being old, a phenomenon that has been termed the "old water paradox" (Kirchner, 2003; McDonnell et al., 2010).

Nevertheless, the fraction of stream water younger than 7 days increased from 1% to up to 45% on an event
scale depending on storm size (Fig. 7b, 8e). This indicated that most precipitation did not mobilize old water in the first place; instead, it drained directly into river networks and contributed to the stream via fast flow paths. This reflects results reported by Szeles et al. (2024), where their findings showed that the new water contribution averaged around 50% during peak flows in selected large events in the HOAL catchment. Given that the slope of the catchment is relatively low at 8%, a possible explanation might be the presence of soil types
with low to moderate permeability and the influence of agricultural land use (Szeles et al., 2024). Another reasons can be the high portion of agriculturally used land which tends to seal at the surface during heavy events, thus inhibiting infiltration.

### 4.4 Catchment Transit times variability with hydrological and hydroclimatic conditions

Based on the assumptions in the model structure and parameters, the resulting fraction of water ages younger
than 7 days and younger than 90 days was more strongly correlated with precipitation intensity than with streamflow or soil moisture for both scenarios. In contrast, older water ages (90 to 365 days) exhibited weak or negative correlations with these hydrological and hydroclimatic drivers (Fig. 7). The fraction of stream water younger than 7 days $F_Q$ (T<7 days), positively correlated with precipitation intensities (Fig. 7b), implying that the volume of event water transmitted to streamflow increases more proportionally with storm size. Similar
results were noted by Szeles et al. (2024), who used hydrograph separation methods and highlighted that new water fractions during events increased with precipitation intensity in the HOAL catchment.


In contrast, the measured volumetric soil water content SWC (%) did not strongly correlate for both scenarios with shorter transit times $F_Q$ (T<7 days) and $F_Q$ (T<90 days) (Fig. 7c, 7f). This may seem contradictory, but it is plausible to assume that the effect of frequent fast flow in the HOAL catchment dominates and masks the

underlying relationship between catchment wetness and transit times. Similar results were found by Hövel et al. (2023) in an analyses of similar event runoff separation. More specifically, they showed that similar runoff responses had stronger correlations with precipitation than measured volumetric soil water content.

Stream water fractions with transit times less than 90 days, $F_Q$ (T<90 days) were weakly correlated with discharge Q (mmd$^{-1}$) (r = 0.40 for S1 and 0.34 for S2) but were strongly positively correlated with precipitation

intensity $P_I$ (mmd$^{-1}$) (r = 0.71 for S1 and 0.62 for S2) (Figs 8d and 8e). The formulation in the model, results in the dominance of fast runoff flow paths and their persistence during both small and large precipitation events in the HOAL catchment. This findings support the earlier study by Freyberg et al. (2018), who noted that low discharge sensitivity to high fractions of young water can indicate the dominance of fast runoff flow paths in the hydrological response. This behavior persists regardless of the magnitude of precipitation events, particularly

under conditions where the landscape promotes rapid water movement, such as in catchments with certain soil types or topographic features (like in the HOAL catchment). Such behavior points also well-developed subsurface flow paths (such as tile drains at the hillslope scale) that efficiently transport water and solutes to the stream, highlighting the catchment's sensitivity to precipitation input.

### 4.5 Implications and limitations

The findings of this paper have important implications for representing transport processes in small, flashy catchments, and for hydrological modelling at large. The application of the model in two different scenarios provided evidence of the critical role of precipitation intensity as an additional dominant control on transit times in the HOAL catchment. Scenario 1 resulted in a higher fraction of water ages $F_Q$ (7<T<90 days) compared to Scenario 2 (Fig. 6b, Table 5) and did not simulate peaks in δ$^{18}$O signals as strongly as Scenario 2. It is

unsurprising that parameterizing the SAS function shape based exclusively on soil moisture results in the shape parameter α(t) being closer to uniform sampling when soil is dry. This formulation, therefore, lead to a higher probability of mobilizing older water (7<T<90 days), rather than the faster transmission of new (T<7 days) water to the streamflow in dry soil condition (Fig. 6b, Table 5). Being conditional on the assumptions made throughout the modeling process, and notwithstanding potential uncertainties, high-frequency water-stable

isotope data and model calibrations provide relatively strong evidence to support the key findings of this study: both soil moisture and precipitation intensity significantly influence hydrological responses and transit times in the HOAL catchment. This led to non-linear flow behavior and a shift toward younger water ages in the stream, particularly during autumn and summer. Soil-wetness-dependent and precipitation-intensity-conditional SAS functions may, therefore, be necessary to better capture and identify the mechanisms driving rapid streamflow

generation and their associated time scales, notably in catchments where preferential flows and overland flow are dominant flow processes.

There are some limitations in this study that need to be addressed and tested in future research. The model calibration based on both scenarios overestimated low flows during the summer of 2016, despite relatively higher precipitation during that year. This overestimation is likely linked to groundwater recharge processes





being more complex than represented in the model structure. The underestimation of low flows began after an occurrence of intense rainfall event (P >50 mmd$^{-1}$, Figure 2) followed by several moderate-intensity events. A potential explanation is the activation of flow paths down to the depth of the tile drainage system or dominant subsurface lateral flow, which may have diverted water directly to the stream, bypassing groundwater infiltration and promoting interflow. Another possibility is the potential presence of a low-permeability unsaturated transition zone between the root zone and the groundwater table which may have delayed groundwater recharge. This could also explain why low flows in the winter and spring of 2017 were conversely underestimated. To fully evaluate these hypotheses and better estimate the recharge processes, additional field observations and more detailed studies focusing on subsurface dynamics and groundwater interactions are necessary.

Furthermore, the model calibration based on both scenarios showed limitations in simulating very low $\delta^{18}$O signals during the summer months, potentially due to the constant value assigned to the division parameter Cn (-) for infiltration-excess overland flows. This parameter was kept constant in this study to maintain model simplicity, as the primary focus was on testing the role of precipitation intensity in water partitioning. However, correlation results with hydrological and hydroclimatic drivers (Fig. 7) suggest that Cn (-) might also be a function of rainfall intensity and could increase with higher precipitation intensities. This indicates the need for a more dynamic representation of Cn to better capture its response to changing rainfall conditions.

Lastly, model calibration resulted in an infiltration-excess overland flow threshold precipitation intensity parameter P$_{tresh}$(mmd$^{-1}$) range between 10–15 (mmd$^{-1}$) and 5–10 (mmd$^{-1}$) for Scenario 1 and Scenario 2, respectively (Fig. S1). While this may seem surprising at first, it can be reasonably explained by surface sealing during rainfall, which inhibits infiltration, particularly in areas affected by agricultural land use in HOAL catchment. Additionally, macropore flow observed in the summer, when the topsoil dries and cracks due to its high clay content, may also contribute to this effect. This parameter was also identified as a threshold for partitioning rainfall into preferential flow pathways and overland flow, promoting fast runoff with minimal interaction with resident water to simulate $\delta^{18}$O signals. Therefore, this threshold should not be considered a definitive marker for infiltration-excess overland flow. Instead, it can be a marker for any processes where the landscape promotes rapid water movement in HOAL catchment.

**5 Conclusion**

In this study, we tested whether fast flow transit times are controlled by soil moisture alone or also by precipitation intensity in an agricultural headwater catchment. The results suggest that both soil moisture and precipitation intensity exert a significant influence on transit times. The data also support the hypothesis that preferential flow age fractions are linearly related to soil moisture when precipitation intensity is below a threshold. However, when precipitation intensity exceeds a threshold, there is a higher probability of new water contributing to fast runoff with little exchange with stored water. The SAS functions based on both soil moisture and precipitation intensity resulted in an increased probability of rapid mobilization of young water F$_Q$ (T<7 days), influenced by precipitation intensity particularly during autumn and summer months. Thus, in catchments where subsurface preferential flow and overland flow dominate, soil moisture-dependent and precipitation





intensity-conditional SAS functions may be required to better the age distribution of quick streamflow. Models that do not account for precipitation may underestimate the impact of intense precipitation events on quick runoff generation in flashy headwater catchments, particularly where infiltration-excess overland flow or rapid
tile drain flow are important runoff mechanisms when the soil is dry.

The findings also underscore the importance of the activation of fast flow paths in water quality variations within the catchment. Estimating young water contributions is essential not only for predicting how contaminants and nutrients are mobilized and transported during hydrological events but also for characterizing the underlying processes that govern the movement and mixing of water through the catchment. The results
presented here focus on a small agricultural headwater catchment with substantial contributions from surface flow and shallow subsurface flow to streamflow. In other catchments with quick subsurface runoff and overland flow, accounting for precipitation in transit times may also better reflect the hydrological dynamics and transport processes and assist in developing effective water management strategies.

*Code and data availability*. An Python script that performs the calculations described in this paper will be deposited in an open-access Github archive repository and the link will be supplied with the final published paper.

*Data availability*. The data used in this study can be obtained from the Austrian Federal Agency for Water Management and upon request.

*Author contributions*. HT and MH jointly developed the model architecture for the catchment. HT performed the analysis presented here and drafted the paper. All authors discussed the design, contributed to the overall concept, and participated in the discussion and writing of the manuscript.

*Competing interests*. Some authors are members of the editorial board of the HESS journal.

*Acknowledgments*. We thank the Austrian Federal Agency for Water Management for providing the data
Petzenkirchen catchment that we used in our analysis.This research is funded by the Austrian Science Fund (FWF–Österreichischer Wissenschaftsfonds) [grant number 10.55776/P34666]. The work of Hatice Turk was supported by the Doctoral School "Human River Systems in the 21st Century (HR21)" of the University of Natural Resources and Life Sciences, Vienna.

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
