# Peer review of "Soil moisture and precipitation intensity jointly control the transit time distribution of quick flow in a flashy headwater catchment"

_Hydrology and Earth System Sciences, 2024_

## Referee Comment (RC1)

This manuscript investigates the influence of soil moisture and precipitation intensity on quick flow transit times in a flashy agricultural catchment. By utilizing δ18O tracer data and enhanced SAS function modeling, the manuscript highlights the importance of accounting for both factors to improve the understanding of quick flow generation and its time scale. While the study has the potential to contribute to the literature, several aspects require improvement to better emphasize the unique contributions and significance of the work.

General Comments:
The abstract needs further refinement to clearly articulate the study's novelty and main findings. Greater emphasis should be placed on the hypothesis of incorporating precipitation intensity into SAS functions and its significance. Additionally, the practical implications of the findings, such as their potential contribution to hydrological modeling and water management strategies, should be explicitly discussed.

In the introduction, the discussion on defining the SAS function shape solely based on soil moisture is limited and does not adequately capture the complexity of hydrological responses in catchments. This section should expand on the rationale for incorporating precipitation intensity as an additional factor. Furthermore, the introduction would benefit from a more structured conclusion that explicitly presents the study's hypothesis and expected outcomes to provide a clearer sense of direction for readers.

The results section does not sufficiently highlight the study's innovative aspects. Greater emphasis should be placed on demonstrating how incorporating precipitation intensity into the SAS function improves tracer simulation and advances the understanding of quick flow processes.

The broader applicability of the findings to different types of catchments should be discussed. Expanding on how the proposed approach could be generalized to catchments with varying hydrological characteristics would enhance the study's relevance and impact.

The conclusions should include a more detailed discussion of the underlying mechanisms and the broader implications of the findings. While the conclusion mentions the potential for contributing to water management strategies, it remains vague. Specific examples of how the results could be applied in practice would make the conclusions more compelling and actionable.

Specific Comments
L33-34: Clarify the meaning of "Flow process promotes contribution of precipitation to the stream." Additional explanation is needed.

Lines 102-109 lack logical clarity, with an abrupt shift from discussing preferential flow in headwater catchments to emphasizing quick flow responses. Refocus the section for better coherence.

L117: The citation is unnecessary in this context.

L149: What does "event intensities" refer to? Are the event intensities specified as 5 mm/h? The phrase "at the end of with that of the following event" is unclear. Please revise for clarity.

L155: Does "Length of event" refer to the event duration? Additionally, how does the "event length" influence the adjustment of sampling frequency? Provide more details.

L254: Check and revise "Equation 2727" as it appears to be an error.

L423: Confirm whether "6.53%" refers to Scenario 2.

L432: Avoid repetition of "based on."

L445: Delete "as" for grammatical accuracy.

L545: The word "stable" is repeated unnecessarily. Remove one instance.

L571: Provide more explanation regarding the mechanism by which "agriculturally used land tends to seal at the surface during heavy events." This statement requires elaboration for better understanding.

L668: The statement "assisting in developing effective water management strategies" is too vague. Include specific examples of potential real-world applications to make this claim more concrete and persuasive.

In addition to addressing the specific comments above, it is recommended to carefully proofread the manuscript to ensure proper grammar, sentence structure, and clarity.

---

## Referee Comment (RC2)

**[General Comments]:**

The manuscript "Soil moisture and precipitation intensity control the transit time distribution of quick flow in a flashy headwater catchment"(ID: hess-2024-359) mainly introduces the influences of both soil moisture and precipitation intensity on transit times, and highlights the rule of precipitation intensity in rapid mobilization of young water using the StorAge Selection (SAS) functions and measured stable isotope data. This work is interesting and significant for the solute-transport model and developing effective water management strategies. But some minor mistakes in this manuscript are found.

Therefore, the article, at current states, needs to be a minor revision, which may be worth publishing for this journal. The following is my comments for further improving the quality of this manuscript.

**[Specific comments]:**

1) The authors calculated the mean and maximum percentage of streamflow fractions for transit times T<7 days, T<90 days, 7<T<90 days, and 90<T<365 days. It is significance for transit times at watershed scales, can you attempt to analyze the rainfall-runoff event in hourly intervals with T<1 day? It is importance to understanding the flood hydrograph.

2) The saturated hydraulic conductivity Ks data should be added to understand the runoff generations.

3) The 4.5 Part-Implications and limitations should be concise.

4) The "conclusion" should be "Conclusions"?

---

## Author Comment (AC1)

Please find below our responses to the comments by the Reviewer. Referee comments are shown in black. Authors replies are in blue

This manuscript investigates the influence of soil moisture and precipitation intensity on quick flow transit times in a flashy agricultural catchment. By utilizing $\delta 18O$ tracer data and enhanced SAS function modeling, the manuscript highlights the importance of accounting for both factors to improve the understanding of quick flow generation and its time scale. While the study has the potential to contribute to the literature, several aspects require improvement to better emphasize the unique contributions and significance of the work.

We thank the reviewer for the positive and constructive feedback, which will help us to improve our work. Below, we outline how we consider the issues raised by the reviewer and the changes we intend to make.

General Comments: The abstract needs further refinement to clearly articulate the study's novelty and main findings. Greater emphasis should be placed on the hypothesis of incorporating precipitation intensity into SAS functions and its significance. Additionally, the practical implications of the findings, such as their potential contribution to hydrological modeling and water management strategies, should be explicitly discussed.

We thank the reviewer for this suggestion. In the revised abstract, we will explicitly highlight the novel aspect of incorporating precipitation intensity into the SAS functions and how this helps refine our understanding of fast flow processes. We will also clarify the significance of this approach for hydrological modeling, particularly in improving transit time estimations and streamflow tracer predictions.

In the introduction, the discussion on defining the SAS function shape solely based on soil moisture is limited and does not adequately capture the complexity of hydrological responses in catchments. This section should expand on the rationale for incorporating precipitation intensity as an additional factor. Furthermore, the introduction would benefit from a more structured conclusion that explicitly presents the study's hypothesis and expected outcomes to provide a clearer sense of direction for readers.

We agree with this comment. We will revise the Introduction to provide a more comprehensive rationale for including precipitation intensity as an additional factor in shaping the SAS functions. Specifically, we will elaborate on how precipitation intensity can influence rapid flow processes (such as infiltration excess, saturation excess overland flow, and agricultural tile drain flow) and its interplay with soil moisture in generating quick flow and shorter transit times. We will also strengthen the concluding paragraph of the Introduction by clearly stating our hypothesis that incorporating precipitation intensity alongside soil moisture for parameterizing time-variable SAS functions will lead to more accurate representations of tracer dynamics in catchments with significant overland flow. Furthermore, we will address the following research questions:

1. *How does formulating the SAS function based solely on soil moisture reflect the tracer simulation in the catchment and the inferred transit times?*

2. *How does formulating the SAS function based on soil moisture conditioned by precipitation intensity affect the tracer simulation in the catchment and the inferred transit times?*

The results section does not sufficiently highlight the study's innovative aspects. Greater emphasis should be placed on demonstrating how incorporating precipitation intensity into the SAS function improves tracer simulation and advances the understanding of quick flow processes.

We appreciate the reviewer's concern regarding the need to highlight the study's innovative aspects in the Results section. We will strengthen this section by explicitly illustrating how tracer simulations for preferential flow differ between the two scenarios. A new figure will be included to demonstrate the model's ability to represent different patterns of $\delta^{18}O$ in simulated preferential flow, thereby highlighting the improvements gained when incorporating precipitation intensity into the SAS function and how this advances our understanding of quick flow processes. Specifically, we will further compare the performance of the precipitation-intensity-based model against the soil-moisture-only model. We will emphasize how these improvements link to specific hydrological processes, such as infiltration-excess overland flow and tile drain flow, thereby explaining why the tracer signal is more aligned with precipitation under high-intensity events.

[Figure]

Figure 1: $\delta^{18}O$ simulations from preferential flow in the model under two scenarios: Scenario 1 (S1, dark blue dots), where the SAS function is based solely on soil moisture, and Scenario 2 (S2, light blue dots), where it additionally accounts for precipitation intensity. In S1, the $\delta^{18}O$ signal is more dampened, indicating a stronger smoothing effect of soil moisture on the tracer input. In contrast, S2 produces $\delta^{18}O$ responses more closely aligned with the precipitation input, reflecting a more direct contribution of event water to streamflow.

The broader applicability of the findings to different types of catchments should be discussed. Expanding on how the proposed approach could be generalized to catchments with varying hydrological characteristics would enhance the study's relevance and impact.

We appreciate this comment and will include a discussion how our proposed approach could be transferred to catchments exhibiting similar hydrological characteristics (other catchments where fast flow is of relevance). Specifically, we will highlight how factors such as land use, and soil properties, influence the role of precipitation intensity in rapid runoff generation. By outlining these considerations, we aim to demonstrate the broader applicability of our findings and how they could inform future studies or modelling strategies in varied catchments

The conclusions should include a more detailed discussion of the underlying mechanisms and the broader implications of the findings. While the conclusion mentions the potential for contributing to water management strategies, it remains vague. Specific examples of how the results could be applied in practice would make the conclusions more compelling and actionable.

We agree with this recommendation highlighting our findings (e.g., how fast flow paths and precipitation intensity interplay drive rapid transit times). We will also highlight specific examples of practical applications such as informing agricultural drainage management, refining flood forecasting tools, and guiding land-use planning to mitigate runoff peaks.

Specific Comments L33-34: Clarify the meaning of "Flow process promotes contribution of precipitation to the stream." Additional explanation is needed.

We agree. We will revise this sentence in the manuscript to clarify that the "flow process facilitates the contribution of precipitation to the stream" refers to rapid flow pathways—such as infiltration-excess overland flow, preferential flow through macropores, and tile drain flow—that allow precipitation water to bypass much of the soil matrix and reach the stream with minimal storage or mixing.

Lines 102-109 lack logical clarity, with an abrupt shift from discussing preferential flow in headwater catchments to emphasizing quick flow responses. Refocus the section for better coherence.

We appreciate this comment and will revise the paragraph for better coherence. Specifically, we will restructure lines 102–109 to create a smoother transition from discussing preferential flow in headwater catchments to highlighting the broader range of rapid flow responses (e.g., infiltration-excess overland flow). We will clarify that while preferential flow is often encapsulated in SAS functions through soil-moisture-dependent mechanisms, rapid runoff can also occur when precipitation intensities exceed the infiltration capacity. This revision will establish a logical progression leading to our hypothesis that incorporating precipitation intensity in SAS functions can improve the representation of tracer dynamics.

L117: The citation is unnecessary in this context.

Thank you for pointing this out. We will remove the citation at Line 117 to maintain conciseness and clarity in the text.

L149: What does "event intensities" refer to? Are the event intensities specified as 5 mm/h? The phrase "at the end of with that of the following event" is unclear. Please revise for clarity.

Thank you for the comment. We will revise the explanation for clarity. The following revision will be made to the manuscript: Once a 0.25, L sampling bottle was filled (corresponding to 5, mm of precipitation), the sampler switched to the next bottle. If the rainfall intensity was too low to fill a bottle within a single event, some precipitation from the end

of one event could mix with precipitation from the start of the following event in the same bottle.

L155: Does "Length of event" refer to the event duration? Additionally, how does the "event length" influence the adjustment of sampling frequency? Provide more details.

Yes, the "length of the event" refers to the event duration. Specifically, to influence the adjustment of sampling frequency for longer events, the sampling could be spaced to cover the full duration without exceeding bottle capacity. We will replace "length" with "duration" in the revised manuscript.

L254: Check and revise "Equation 2727" as it appears to be an error.

Thank you for pointing this out. We will correct "Equation 2727" to its intended reference "Equation 27".

L423: Confirm whether "6.53%" refers to Scenario 2.

Yes, the value "6.53%" refers to Scenario 2. We will revise the text to make this reference clearer.

L432: Avoid repetition of "based on."

We will revise the sentence at Line 432 to remove the repeated phrase "based on,"

L445: Delete "as" for grammatical accuracy.

We will delete "as" for grammatical accuracy

L545: The word "stable" is repeated unnecessarily. Remove one instance.

Thank you for highlighting this repetition. We will remove the extra instance of the word "stable" to improve readability and clarity.

L571: Provide more explanation regarding the mechanism by which "agriculturally used land tends to seal at the surface during heavy events." This statement requires elaboration for better understanding.

We will clarify that agricultural soils, particularly those with intensive tillage or limited vegetative cover are prone to surface sealing (Laloy and Bielders, 2010) during heavy rainfall events (e.g a reduction in infiltration capacity from heavy machinery, soil erosion processes). The impact of raindrops can break down soil aggregates at the surface, leading to the formation of a thin, compacted soil layer (i.e., a "soil crust") that reduces infiltration capacity (Blöschl, 2022). This crust formation can be more pronounced in soils with higher silt content or weak aggregate stability, thereby favoring overland flow and quick runoff generation during heavy precipitation.

L668: The statement "assisting in developing effective water management strategies" is too vague. Include specific examples of potential real-world applications to make this claim more concrete and persuasive.

In the revised manuscript, we will provide examples of how these findings can inform effective water management strategies in agricultural catchments. For instance, by pinpointing periods or conditions under which precipitation intensity triggers rapid flow through tile drains and preferential pathways, managers could better schedule fertilizer applications to minimize nutrient leaching and reduce water quality deterioration. Similarly, insights into quick flow processes can guide the placement and timing of agricultural drainage systems to mitigate peak flows, inform stormwater management interventions (e.g., retention ponds or buffer strips) to reduce runoff peaks, and land-use practices to prevent or minimize the direct contribution of event water to streams. These practical examples will underscore how our improved understanding of rapid flow pathways can lead to tangible benefits in real-world water management scenarios.

In addition to addressing the specific comments above, it is recommended to carefully proofread the manuscript to ensure proper grammar, sentence structure, and clarity.

We fully acknowledge the importance of clear and concise writing. As suggested, we will meticulously proofread the manuscript to address any grammatical or structural issues and to enhance the overall clarity of the text.

**References**

Blöschl, G.: Flood generation: process patterns from the raindrop to the ocean, Hydrology and Earth System Sciences, 26, 2469–2480, 2022.

Laloy, E. and Bielders, C. L.: Effect of intercropping period management on runoff and erosion in a maize cropping system, Journal of environmental quality, 39, 1001–1008, 2010.

---

## Author Comment (AC2)

Please find below our responses to the comments by the Reviewer. Referee comments are shown in black. Authors replies are in blue

[General Comments]: The manuscript "Soil moisture and precipitation intensity control the transit time distribution of quick flow in a flashy headwater catchment"(ID: hess-2024-359) mainly introduces the influences of both soil moisture and precipitation intensity on transit times, and highlights the rule of precipitation intensity in rapid mobilization of young water using the StorAge Selection (SAS) functions and measured stable isotope data. This work is interesting and significant for the solute-transport model and developing effective water management strategies. But some minor mistakes in this manuscript are found. Therefore, the article, at current states, needs to be a minor revision, which may be worth publishing for this journal. The following is my comments for further improving the quality of this manuscript.

We are grateful to the reviewer for taking the time to read the manuscript and for the positive evaluation of our work.

[Specific comments]: 1) The authors calculated the mean and maximum percentage of streamflow fractions for transit times T<7 days, T<90 days, 7<T<90 days, and 90<T<365 days. It is significance for transit times at watershed scales, can you attempt to analyze the rainfall-runoff event in hourly intervals with T<1 day? It is importance to understanding the flood hydrograph.

We appreciate the reviewer's suggestion to analyze transit times at an hourly resolution (i.e., for T < 1 day) to capture the flood hydrograph better. In principle, this would indeed offer valuable insights into sub-daily dynamics. However, our tracer dataset does not currently support modeling at an hourly time step due to its sampling resolution. Nonetheless, we acknowledge the importance of understanding very short transit times, particularly for flood events. To address this point, we propose to include a summary of the fraction of streamflow with T < 2 days in our results. This can serve as a proxy for the 1-day threshold within the constraints of our data. Additionally, we will discuss in the manuscript how event-based hydrograph separation methods could complement our approach to further elucidate rapid flow pathways during intense rainfall events.

2) The saturated hydraulic conductivity Ks data should be added to understand the runoff generations.

We understand this comment. We will add the saturated hydraulic conductivity (Ks) information to the study site description section

3) The 4.5 Part-Implications and limitations should be concise.

We agree with the reviewer's suggestion. We will revise Section 4.5 to present the implications and limitations of our study in a concise manner, focusing on the key take-home

messages, practical benefits for hydrological modeling, and clear acknowledgment of the main constraints in our approach.

4) The "conclusion" should be "Conclusions"?

We agree. We will change this to "Conclusions"

---

## Author Response (AR1)

Public justification (visible to the public if the article is accepted and published):

Based on the referees' comments and the authors' responses, I believe the revised manuscript will make a significant contribution to the runoff generation mechanism literature, offering substantial new findings. Please submit the revised manuscript, ensuring that all comments are addressed thoroughly. I will then initiate another round of review.

Additionally, please ensure the language is polished before submission.

Thank you.

**Point-by-Point Authors' Responses to the Editor and Reviewers Comments.-**

**Authors' General Comment:** We are very grateful to the Editor and to the reviewers for their valuable comments. All scientific and technical suggestions provided have strengthened the revised version of our manuscript. We appreciate the time and effort invested in reviewing our manuscript and providing feedback, which has helped us improve the scientific clarity and presentation of our work. We addressed all comments, and we believe the revised version of the manuscript has been substantially improved as a result.

In the following, we provide our detailed **point-by-point** responses to each of the comments. Reviewer comments are shown in **black**, while our responses are shown in **blue**. Line numbers refer to the revised version with track-changes.

**Response to Reviewer # 1**

**General Comment:** This manuscript investigates the influence of soil moisture and precipitation intensity on quick flow transit times in a flashy agricultural catchment. By utilizing δ$^{18}$O tracer data and enhanced SAS function modeling, the manuscript highlights the importance of accounting for both factors to improve the understanding of quick flow generation and its time scale. While the study has the potential to contribute to the literature, several aspects require improvement to better emphasize the unique contributions and significance of the work.

**Reply:** We thank the reviewer for the positive and constructive feedback, which helped to improve our work. Below, we address the issues raised by the reviewer and outline the changes we made.

**Comment:** The abstract needs further refinement to clearly articulate the study's novelty and main findings. Greater emphasis should be placed on the hypothesis of incorporating precipitation intensity into SAS functions and its significance. Additionally, the practical implications of the findings, such as their potential contribution to hydrological modeling and water management strategies, should be explicitly discussed.

**Reply:** We appreciate this suggestion and revised the abstract to more clearly highlight the study's novel contributions and key results. Specifically, we now emphasize the hypothesis that "in a heterogeneous catchment with a significant fast runoff response component, precipitation intensity, in addition to soil moisture, plays a critical role in the preferential release of younger water". We also discuss how these findings can inform hydrological modelling approaches and guide practical water management strategies, particularly in small, flashy catchments where precipitation intensity is a dominant control. (Lines 21-23 and 33-36).

"Thus, in catchments where preferential flows and overland flow are dominant flow processes, soil-wetness-dependent and precipitation-intensity-conditional SAS functions may be required to better describe the time scale of solute transport in modelling, which has implications for stream water quality and agricultural management practices such as the timing of fertilizer application."

**Comment:** In the introduction, the discussion on defining the SAS function shape solely based on soil moisture is limited and does not adequately capture the complexity of hydrological responses in catchments. This section should expand on the rationale for incorporating precipitation intensity as an additional factor. Furthermore, the introduction would benefit from a more structured conclusion that explicitly presents the study's hypothesis and expected outcomes to provide a clearer sense of direction for readers.

**Reply:** We revised the introduction to underscore the complexity of hydrological responses and to clarify why incorporating precipitation intensity is essential (Lines 87–95). Specifically, we expanded the discussion on the limitations of relying solely on soil moisture to define the SAS function shape, highlighting that this approach may overlook non-linear processes and the potential for fast runoff generation in flashy catchments. We added:

(Lines 88-98)

"Although preferential flow is often accounted for in SAS functions through soil-moisture-dependent mechanisms, rapid runoff can also occur independently of the soil-moisture state when precipitation intensities exceed infiltration capacity. This is particularly critical for catchments where flow generation is not linearly related to soil storage, such as the flashy Weierbach catchment in Luxembourg (Rodriguez and Klaus, 2019), and for catchments where precipitation intensity and duration may play a critical role in how quickly water is mobilised from the landscape due to moderate to low soil infiltration capacity, such as the Hydrological Open-Air Laboratory in Austria (Vreugdenhil et al., 2022). Exclusively basing the shape of the SAS function on soil moisture may not fully capture the complexity of hydrological responses or all relevant transport processes, due to non-linear relationships between storage and streamflow (Danesh-Yazdi et al., 2018; Rodriguez & Klaus, 2019). Therefore, it remains to be tested whether accounting for precipitation intensity in addition to soil moisture to parameterize time-variable SAS functions may yield improved representations of stream tracer dynamics in specific environments."

We conclude this expanded section by explicitly stating our hypothesis (Lines 99–110):

"The objective of this study is to test two alternative approaches to formulate the shape of time-variable SAS functions to account for the higher probability of releasing young water in a flashy headwater catchment: (i) exclusively soil moisture controls the SAS function shape for preferential release of younger water, and (ii) soil moisture and precipitation intensity jointly control the SAS function shape for preferential release of younger water. We hypothesize that in a heterogeneous catchment with a significant fast runoff response component, precipitation intensity, in addition to soil moisture, plays a critical role in the preferential release of younger water to the stream. To test this hypothesis, we used high-resolution $\delta^{18}O$ data (weekly and event-based streamflow $\delta^{18}O$ samples) in the 66 ha agricultural catchment Petzenkirchen (Hydrological Open Air Laboratory, HOAL) in Austria. We addressed the following questions.

1. How does formulating the SAS function based solely on soil moisture reflect the streamflow tracer simulation and the inferred transit times?

2. How does formulating the SAS function based on soil moisture conditioned by precipitation intensity affect the streamflow tracer simulation in the catchment and the inferred transit times?"

**Comment:** The results section does not sufficiently highlight the study's innovative aspects. Greater emphasis should be placed on demonstrating how incorporating precipitation intensity into the SAS function improves tracer simulation and advances the understanding of quick flow processes.

**Reply:**

We appreciate the reviewer's concern regarding the need to highlight the study's innovative aspects in the results section. In the revised manuscript (Lines 385–398), we strengthened this section by explicitly explaining how tracer simulations for preferential flow differ between the two scenarios and added a new figure (Figure 6) to highlight the difference. This figure demonstrates the model's ability to represent different patterns of $\delta^{18}O$ in simulated preferential flow, thereby highlighting the improvements gained when incorporating precipitation intensity into the SAS function and how this advances our understanding of quick flow processes (Lines 386-390): At Lines 391–398, we now include the following text and figure in the Results section:

"Simulated $\delta^{18}O$ from preferential flow (RF) is shown in Figure 6. In Scenario 1, where the SAS function for preferential flow is exclusively controlled by soil moisture, the simulated $\delta^{18}O$ signal in stream water was noticeably more dampened compared to Scenario 2. This reflects a higher probability of mobilising older stored water, reducing the direct transmission of event water to the stream. In contrast, Scenario 2, which incorporated both soil moisture and precipitation intensity, resulted in a modelled stream water $\delta^{18}O$ signal that more closely resembled both, the observed stream water $\delta^{18}O$ as well as the precipitation input signal. This highlights that accounting for precipitation intensity allows the model to better capture dynamic preferential flow responses, with a higher probability of mobilising younger event water into the stream."

[Figure]

**Figure 6:** $\delta^{18}O$ simulations of preferential flow under two scenarios: Scenario 1 (S1, dark blue dots), where the SAS function was based solely on soil moisture, and Scenario 2 (S2, light blue dots), where it additionally accounted for precipitation intensity. In S1, the $\delta^{18}O$ signal was more dampened, indicating a larger contribution of old water to preferential flow. In contrast, S2's $\delta^{18}O$ response more closely aligned with the precipitation input, reflecting a more direct contribution of event water to streamflow.(b) zoom-in to the $\delta18O$ simulations of preferential flow under two scenarios for the year 2016.

As an additional note, we also updated Figure 5e as we found a small error in plotting the NSE values for Scenario 2.

**Comment:** The broader applicability of the findings to different types of catchments should be discussed. Expanding on how the proposed approach could be generalized to catchments with varying hydrological characteristics would enhance the study's relevance and impact.

**Reply:**

We appreciate this comment and added a discussion on how our proposed approach could be applied to catchments with diverse hydrological characteristics—particularly those where fast flow processes dominate. As noted in Lines [580–585]:

"Although the analysis is here limited to a small, agricultural catchment with flashy response, it is plausible to assume that the approach also is similarly valid in other settings with diverse hydrological characteristics and where rainfall intensity exceeds infiltration rates, leading to surface runoff or subsurface preferential pathways. By focusing on how soil moisture and precipitation intensity jointly influence younger water release, the insights of this study can help to develop water management strategies in, e.g., agricultural catchments.
Managers could better schedule fertilizer applications to minimize nutrient leaching and reduce water quality deterioration."

**Comment:** The conclusions should include a more detailed discussion of the underlying mechanisms and the broader implications of the findings. While the conclusion mentions the potential for contributing to water management strategies, it remains vague. Specific examples of how the results could be applied in practice would make the conclusions more compelling and actionable.

**Reply:** We appreciate this feedback and revised the conclusions section to include more explicit examples of how our findings can help to develop water management strategies (Lines 625–634). In particular, we now note:

"For instance, by identifying periods or conditions under which precipitation intensity triggers rapid flow through tile drains and preferential pathways, will allow watermanagers to develop guidance for better fertilizer application schedules to minimize nutrient export and reduce water quality deterioration. Similarly, insights into quick flow processes can guide the placement and timing of agricultural drainage systems to mitigate peak flows, inform stormwater management interventions (e.g., retention ponds or buffer strips) to reduce runoff peaks,
and land-use practices to prevent or minimize the direct contribution. "

**Specific Comments:**
**Comment (Lines 33-34):** Clarify the meaning of "Flow process promotes contribution of precipitation to the stream." Additional explanation is needed.

Thank you for noting this point. We revised the sentence at (Lines 30–32), we now note:

"rapid flow pathways such as infiltration-excess overland flow, preferential flow through macropores, and tile drain flow—allowing precipitation water to bypass much of the soil matrix and reach the stream with minimal storage or mixing, even under dry soil conditions."

**Comment (Lines 102–109):** Lack logical clarity, with an abrupt shift from discussing preferential flow in headwater catchments to emphasising quick flow responses. Refocus the section for better coherence.

**Reply:** We appreciate this feedback and revised the paragraph to ensure better coherence (Lines 87–90).
"Although preferential flow is often accounted for in SAS functions through soil‑moisture‑dependent mechanisms, rapid runoff can also occur independently of the soil-moisture state when precipitation intensities exceed infiltration capacity.
(Lines 86–98) Therefore, it remains to be tested whether accounting for precipitation intensity in addition to soil moisture to parameterise time-variable SAS functions may yield improved representations of stream tracer dynamics in specific environments.

**Comment (Line 117):** The citation is unnecessary in this context.

**Reply:** We removed the citation at Line 117

**Comment (Line 149):** What does "event intensities" refer to? Are the event intensities specified as 5 mm/h? The phrase "at the end of with that of the following event" is unclear. Please revise for clarity.

**Reply:** We revised the section for clarity by specifying the sampling criteria and how low rainfall can lead to a mixture of precipitation from consecutive events: (Lines 146–150)

"Once a 0.25 L sampling bottle was filled (corresponding to 5 mm of precipitation), the sampler switched to the next bottle. If the rainfall amount was too low to fill a bottle within a single event, some precipitation from the end of one event could mix with precipitation from the start of the following event in the same bottle."

**Comment (Line 155):** Does "Length of event" refer to the event duration? Additionally, how does the "event length" influence the adjustment of sampling frequency? Provide more details.

    **Reply:** We replaced "length" with "duration" in the revised manuscript, (Lines 154–155), to clarify that this refers to the time span of an event. The duration of a runoff event influenced the sampling frequency of the Isco 6712 sampler as the aim was to evenly spread out the the 24 bottles of 1 L each. For example, should the sampling frequency be too high (e.g., 15 min) but the event last for 20 hours, then only the first part of the rising limb would be sampled, and information about the recession of the event would be lost. We added to the 135     manuscript that an even distribution of sampling bottles was aimed for with adjusting the sampling frequency to the anticipated event duration.

    **Comment (Line 254):** Check and revise "Equation 2727" as it appears to be an error.

    **Reply:** We corrected "Equation 2727" to "Equation 27" as intended.

    **Comment (Line 423):** Confirm whether "6.53%" refers to Scenario 2.

    **Reply:** We revised the text to that "6.53%" refers to Scenario 2.

    **Comment (Line 432):** Avoid repetition of "based on."

    **Reply:** We removed the repeated phrase "based on"

    **Comment (Line 445):** Delete "as" for grammatical accuracy.

**Reply:** We deleted "as" for grammatical accuracy

    **Comment (Line545):** The word "stable" is repeated unnecessarily. Remove one instance.

    **Reply:** We deleted "stable" for grammatical accuracy

**Comment (Line 571):** Provide more explanation regarding the mechanism by which "agriculturally used land tends to seal at the surface during heavy events." This statement requires elaboration for better understanding.

**Reply:** We expanded the discussion in the revised manuscript (Lines 542–549) to clarify the mechanism of surface sealing in agricultural soils:

"Another reason can be the high proportion of agriculturally used land, which tends to develop a compacted or sealed surface layer thus inhibiting or reducing infiltration during heavy rainfall events. Agricultural soils, particularly those with intensive tillage or limited vegetative cover are prone to surface sealing (Laloy and Bielders, 2010) during heavy rainfall events (e.g a reduction in infiltration capacity from heavy machinery, soil erosion processes). The impact of raindrops can break down soil aggregates at the surface, leading to the formation of a thin, compacted soil layer (i.e., a "soil crust") that reduces infiltration capacity (Blöschl, 2022). This crust formation can be more pronounced in soils with higher silt content or weak aggregate stability, thereby favouring overland flow and quick runoff generation during heavy precipitation."

**Comment (Line 432):** The statement "assisting in developing effective water management strategies" is too vague. Include specific examples of potential real-world applications to make this claim more concrete and persuasive.

**Reply:** We revised the text (Lines 627–633) to include explicit examples of how our findings can be applied to real-world water management strategies:

" For instance, by identifying periods or conditions under which precipitation intensity triggers rapid flow through tile drains and preferential pathways, will allow watermanagers to develop guidance for better fertilizer application schedules to minimize nutrient export and reduce water quality deterioration. Similarly, insights into quick flow processes can guide the placement and timing of agricultural drainage systems to mitigate peak flows, inform stormwater management interventions (e.g., retention ponds or buffer strips) to reduce runoff peaks, and land-use practices to prevent or minimize the direct contribution.e runoff peaks, and land-use practices to prevent or minimize the direct contribution."

**Comment:** In addition to addressing the specific comments above, it is recommended to carefully proofread the manuscript to ensure proper grammar, sentence structure, and clarity.
We acknowledge this recommendation and conducted a comprehensive proofreading of the entire manuscript. Particular attention was given to grammar, sentence structure, and overall clarity

**Response to reviewer# 2 :**

**General Comments:** The manuscript "Soil moisture and precipitation intensity control the transit time distribution of quick flow in a flashy headwater catchment"(ID: hess-2024-359) mainly introduces the influences of both soil moisture and precipitation intensity on transit times, and highlights the rule of precipitation intensity in rapid mobilization of young water using the StorAge Selection (SAS) functions and measured stable isotope data. This work is interesting and significant for the solute-transport model and developing effective water management strategies. But some minor mistakes in this manuscript are found. Therefore, the article, at current states, needs to be a minor revision, which may be worth publishing for this journal. The following is my comments for further improving the quality of this manuscript.:
reply:
Reply: We are grateful to the reviewer for taking the time to read the manuscript and for the positive evaluation of our work.

**Specific comments: 1)** The authors calculated the mean and maximum percentage of streamflow fractions for transit times T<7 days, T<90 days, 7<T<90 days, and 90<T<365 days. It is significance for transit times at watershed scales, can you attempt to analyze the rainfall-runoff event in hourly intervals with T<1 day? It is importance to understanding the flood hydrograph.

**Reply:** We appreciate the reviewer's valuable suggestion highlighting the importance of understanding very short transit times, especially in the context of flood hydrographs. To address this, we now included an additional summary of the fraction of streamflow with **T < 2 days** in our results section (see Table 4 Lines 402–405). However, due to the daily interval of our data, it is not possible to analyze transit times <1 day on an hourly time scale.

**Specific comments: 2)** The saturated hydraulic conductivity Ks data should be added to understand the runoff generations.

**Reply:**

We agree with the reviewer that **K**s data provide important context for understanding runoff generation processes in the catchment. To address this, we added the following description to the **site description section** (see Lines 122–126):

"Previous study in the catchment showed that the saturated hydraulic conductivity, Ks, exhibits substantial variability despite relatively limited spatial variation in physical and topographical soil properties. In arable fields, median Ks is approximately three times higher than in grassland areas, with arable land having a mean Ks of around 47 mm h⁻¹ compared to 20 mm h⁻¹ in grassland. Overall, Ks ranges over two orders of magnitude, from as low as 1 mm h⁻¹ to as high as 130 mm h⁻¹, with a coefficient of variation (CV) of around 75 % in arable land (Picciafuoco et al., 2019)."

**Specific comments: 3)** The 4.5 Part-Implications and limitations should be concise.

**Reply:** We revised **Section 4.5** to make it more concise and focused (see Lines 572–586).

We updated the implication as:

"Being conditional on the assumptions made throughout the modelling process, and notwithstanding potential uncertainties, high-frequency water-stable isotope data and model calibrations provide relatively strong evidence to support the key findings of this study: both soil moisture and precipitation intensity significantly influence hydrological responses and transit times in the HOAL catchment. This reflects non-linear flow behaviour and a shift towards younger water ages in the stream, particularly during autumn and summer. Soil- wetness-dependent and precipitation-intensity-conditional SAS functions may, therefore, be necessary to better capture and identify the mechanisms driving rapid streamflow generation and their associated timescales, notably in catchments where preferential flows and overland flow are dominant flow processes. The SAS functions based on both soil moisture and precipitation intensity resulted in an increased probability of rapid mobilization of young water which is critical for stream water quality and groundwater recharge.

Although the analysis is here limited to a small, agricultural catchment with flashy response, it is plausible to assume that the approach also is similarly valid in other settings with diverse hydrological characteristics and where rainfall intensity  exceeds infiltration rates, leading to surface runoff or subsurface preferential pathways. By focusing on how soil moisture and precipitation intensity jointly influence younger water release, the insights of this study can help to develop water management strategies in, e.g., agricultural catchments. Managers could better schedule fertilizer applications to minimize nutrient leaching and reduce water quality deterioration.

**Specific comments: 4)** The "conclusion" should be "Conclusions"?

**Reply:** We corrected the heading to **"Conclusions"** in the revised manuscript (Line XX).

**References**

Blöschl, G.: Flood generation: process patterns from the raindrop to the ocean, Hydrol. Earth Syst. Sci., 26, 2469–2480, https://doi.org/10.5194/hess-26-2469-2022, 2022.

Laloy, E., & Bielders, C. L. (2010). Effect of intercropping period management on runoff and erosion in a maize cropping system. *Journal of environmental quality*, *39*(3), 1001-1008.

https://doi.org/10.2134/jeq2009.0239

Picciafuoco, T., Morbidelli, R., Flammini, A., Saltalippi, C., Corradini, C., Strauss, P., & Blöschl, G. (2019). On the estimation of spatially representative plot scale saturated hydraulic conductivity in an agricultural setting. *Journal of hydrology*, *570*, 106-117.https://doi.org/10.1016/j.jhydrol.2018.12.044

---

## Author Response (AR2)

**Editor Comments**

Public justification (visible to the public if the article is accepted and published):

Dear Authors,

5  Thank you for your thorough revision. All comments from the referees have been reasonably addressed. I am happy to accept the paper for final publication, but please make a technical correction based on the remaining comment from Referee#1.

Thank you again for considering HESS for your research outcome.

10          **Responses to the Editor and Reviewers Comments.-**

**Authors' General Comment:** We sincerely thank the Editor for accepting our manuscript for publication in Hydrology and Earth System Sciences, and we are grateful to the reviewers for their constructive and thoughtful feedback throughout the review process. We have carefully addressed the remaining technical correction and believe the manuscript is now fully

15  ready for final publication.

Reviewer #1 – Final Comment1:

"The captions of several figures and tables (e.g., Figure 3 and Table 1) are overly long and could benefit from refinement. Consider moving some of the descriptive content into the main text to improve readability. "

20  Thank you for this helpful suggestion. We have shortened the captions of Figures and Tables to improve readability and moved the more detailed descriptive content into the main text. (Lines 162 - 168, 178-182, 382-385, and 386 -388)

Reviewer #1 – Final Comment2:

"In Table 2, please check the expression "[7.39e-6, 4.06e-06" – it appears that the closing bracket is missing."

25

Response:

This has been corrected. The missing closing bracket has been added to the expression in Table 2.

**Additional Editorial Corrections:**

30  The institutional affiliation has been updated to reflect the recent change.(Lines 5-7)

The acknowledgement section has been revised in accordance with updated funding requirements. (Lines 636-638)

Kind regards,